# StretchTime: Adaptive Time Series Forecasting via Symplectic Attention

**Yubin Kim**[1]  **Viresh Pati**[1]  **Jevon Twitty**[1]  **Vinh Pham**[1]
**Shihao Yang**[1]  **Jiecheng Lu**[1]

## Abstract

Transformer architectures have established strong baselines in time series forecasting, yet they typically rely on positional encodings that assume uniform, index-based temporal progression. However, real-world systems, from shifting financial cycles to elastic biological rhythms, frequently exhibit "time-warped" dynamics where the effective flow of time decouples from the sampling index. In this work, we first formalize this misalignment and prove that rotary position embedding (RoPE) is mathematically incapable of representing non-affine temporal warping. To address this, we propose Symplectic Positional Embeddings (SyPE), a learnable encoding framework derived from Hamiltonian mechanics. SyPE strictly generalizes RoPE by extending the rotation group $SO(2)$ to the symplectic group $Sp(2, \mathbb{R})$, modulated by a novel input-dependent adaptive warp module. By allowing the attention mechanism to adaptively dilate or contract temporal coordinates end-to-end, our approach captures locally varying periodicities without requiring pre-defined warping functions. We implement this mechanism in StretchTime, a multivariate forecasting architecture that achieves state-of-the-art performance on standard benchmarks, demonstrating superior robustness on datasets exhibiting non-stationary temporal dynamics.[1]

## 1. Introduction

Time series forecasting is fundamental across domains including finance, healthcare, climate science, and industrial monitoring. Transformer architectures have demonstrated

[1]Georgia Institute of Technology, Atlanta, GA, USA. Correspondence to: Shihao Yang <shihao.yang@isye.gatech.edu>, Jiecheng Lu <jliu414@gatech.edu>.

*Proceedings of the 43rd International Conference on Machine Learning*, Seoul, South Korea. PMLR 306, 2026. Copyright 2026 by the author(s).

[1]Code available at https://github.com/yubinkim04/StretchTime

remarkable success in modeling sequential data (Vaswani et al., 2017), and recent adaptations for time series (Zhou et al., 2021; Wu et al., 2021; Nie et al., 2023; Liu et al., 2024) have established strong baselines on standard benchmarks. However, a critical question remains underexplored: how should temporal structure be encoded to capture the time-warped temporal dynamics inherent in real-world sequences?

Positional encodings are central to the Transformer's ability to model sequential dependencies. The original sinusoidal encodings (Vaswani et al., 2017) and their successors, Rotary Position Embeddings (RoPE) (Su et al., 2024) and ALiBi (Press et al., 2022), encode position through fixed functions of token indices. While effective for natural language, these methods assume uniform, index-based temporal spacing, an assumption that fails when the underlying dynamics exhibit *temporal stretching* (Figure 1): time-varying periodicity arising from sensor drift, evolving seasonal patterns, or irregular sampling.

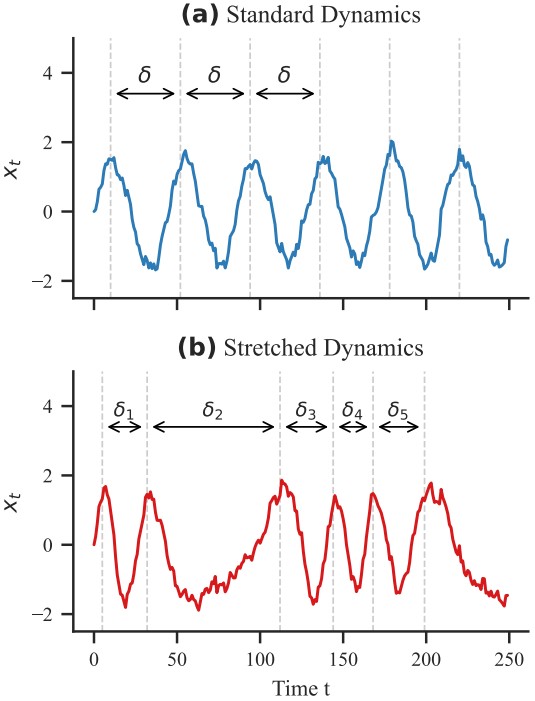

**(a)** Standard Dynamics

**(b)** Stretched Dynamics

*Figure 1.* Visualization of Temporal Stretching

Consider a seasonal autoregressive process whose period $P(t)$ varies over time. Such non-stationary periodicity is ubiquitous in real-world systems: in macroeconomics, financial cycles exhibit state-dependent durations that decouple from standard business cycles (Borio, 2014), while in ecology, climate change drives non-uniform shifts in phenological events across trophic levels (Thackeray et al., 2016). Classical SARIMA models (Box et al., 2015) and their periodic extensions (Franses & Paap, 2004) assume fixed seasonal lags, while dynamic time warping (Sakoe & Chiba, 1978) can align sequences post-hoc but does not integrate with end-to-end learning. In self-attention, the attention logit is dominated by the content-dependent query–key interaction $\langle \mathbf{q}(\mathbf{x}_m), \mathbf{k}(\mathbf{x}_n) \rangle$, and RoPE effectively assumes that this content geometry is stable across time so that a fixed, index-based phase modulation suffices. Fixed-frequency positional encodings face a fundamental limitation: they cannot adapt their attention patterns to match locally varying periodicity, forcing the model to approximate time-varying structure with position-independent frequencies.

We propose **Symplectic Positional Embeddings (SyPE)**, a learnable temporal encoding that strictly generalizes RoPE by extending the rotation group $\mathrm{SO}(2)$ to the symplectic group $\mathrm{Sp}(2, \mathbb{R})$. This extension preserves RoPE's beneficial properties–relative position encoding, length generalization, and computational efficiency–while introducing learnable parameters that enable adaptive temporal transformations including anisotropic scaling and frequency modulation. Concretely, we introduce (i) an *adaptive warp module* that maps content to input-dependent temporal coordinates and (ii) a *learnable rotation kernel* whose symplectic parameters modulate the effective phase and geometry of the positional transformation. We integrate these components into StretchTime, a novel multivariate forecasting architecture designed to learn these warped dynamics end-to-end alongside the forecasting task.

Our contributions are the following:

1. We formalize the temporal stretching problem in time series forecasting and demonstrate the fundamental limitation of fixed-frequency positional encodings for sequences with time-varying periodicity.

2. We derive SyPE from first principles using Hamiltonian mechanics, proving that it strictly generalizes RoPE while maintaining symplectic structure and relative position encoding properties.

3. We implement StretchTime, a Transformer-based forecasting model utilizing SyPE, and demonstrate state-of-the-art performance on standard benchmarks, showing particular improvements on datasets exhibiting non-uniform temporal dynamics.

## 2. Related Work

### 2.1. Transformers for Time Series

The application of Transformers to time series forecasting has evolved rapidly (Wen et al., 2023). Early work like Informer (Zhou et al., 2021) and Autoformer (Wu et al., 2021) addressed complexity via sparse attention and decomposition. Later, PatchTST (Nie et al., 2023) utilized patching for efficiency, while iTransformer (Liu et al., 2024) inverted the framework to treat variates as tokens.

However, the necessity of such complexity is debated. Zeng et al. (2023) demonstrated that simple linear models (DLinear) often match Transformers (Liu et al., 2024), and TimesNet (Wu et al., 2023) achieved state-of-the-art results using 2D convolutions. These findings suggest that capturing temporal structure effectively is more critical than architectural depth, motivating our focus on learnable temporal encodings.

### 2.2. Positional Encodings

The original Transformer (Vaswani et al., 2017) utilized sinusoidal encodings with fixed frequencies to represent absolute positions. While foundational, this approach often fails to generalize when models are evaluated on sequence lengths exceeding those seen during training (Press et al., 2022). To address this, attention shifted toward relative position encodings, which define position based on the distance between tokens rather than their absolute coordinates.

Approaches like T5 (Raffel et al., 2020) introduce learned scalars to bias attention based on token distance, whereas ALiBi (Press et al., 2022) subtracts a static linear penalty, allowing for extrapolation without learned parameters. However, both methods assume that positional relationships are independent of the input content. Recent work suggests that no single encoding dominates across all tasks (Kazemnejad et al., 2023), implying that optimal temporal structure is task-dependent—motivating our learnable approach.

**Rotary Position Embeddings (RoPE)** (Su et al., 2024) represent the current standard in Large Language Models. RoPE encodes relative position by rotating query and key vectors in paired dimensions, providing multi-scale resolution via geometric progression. While effective in other domains (Heo et al., 2024; Chen et al., 2023), RoPE relies on fixed, pre-computed frequencies. This rigidity prevents the model from adapting to the locally varying periodicities inherent in time series—a limitation we formally prove in Theorem 3.1. We generalize RoPE via symplectic transformations, enabling the model to adaptively encode task-specific periodicities.

### 2.3. Time-Varying Periodicity in Real-World Systems

Strict periodicity is rare; instead, systems exhibit "elastic" dynamics. In biomedical monitoring, ECG signals display continuous warping of heartbeat intervals (McCraty & Shaffer, 2015), necessitating non-linear alignment. This elasticity extends to neuroscience (Wiafe et al., 2024) and psychiatry, where symptom trajectories unfold at heterogeneous speeds that obscure dynamics under fixed-time assumptions (Kopland & Giltay, 2025).

This phenomenon extends to macro-systems. In economics, Borio (2014) establishes that financial cycles operate on "elastic" timelines (16–20 years) that decouple from the standard business cycle. Likewise, in ecology, climate change drives differential temporal shifts across trophic levels, causing systemic desynchronization (Thackeray et al., 2016). These findings underscore that temporal warping is heterogeneous and motivate a learnable, adaptive approach to temporal encoding.

### 2.4. Time-Varying Periodicity in Classical Statistics

Modeling time-varying periodicity has a long statistical history. Periodic autoregressive (PAR) models (Franses & Paap, 2004; Basawa & Lund, 2001) allow coefficients to vary seasonally, but assume the underlying period is fixed.

For truly variable pacing, Dynamic Time Warping (DTW) (Sakoe & Chiba, 1978; Berndt & Clifford, 1994) provides non-parametric alignment. DTW finds an optimal path to stretch or compress time, matching patterns across sequences. While effective for post-hoc analysis (Keogh & Ratanamahatana, 2005), it lacks gradient-based integration and requires full sequences at inference.

State-space approaches like TBATS (De Livera et al., 2011) capture evolving dynamics via time-varying coefficients, but rely on rigid parametric forms (Koopman & Ooms, 2006). Our adaptive warp module acts as a differentiable neural analogue to time warping. Unlike post-hoc alignment, it learns to adaptively dilate or contract temporal coordinates end-to-end, without requiring fixed warping function specifications.

## 3. Methodology

### 3.1. Task, Notation, and Architectural Overview

We consider the multivariate time series forecasting problem. Let the observed context be $\mathbf{X} \in \mathbb{R}^{L \times C}$, where $L$ is the lookback window length and $C$ is the number of input channels. Given a forecast horizon $T$, the goal is to predict the future values $\mathbf{Y} \in \mathbb{R}^{T \times C}$. For simplicity in notation, we omit the batch dimension $B$ unless necessary.

Our model operates on a concatenated sequence of length $N = L + T$. For each target channel, we construct a token sequence $\mathbf{H} = [\mathbf{h}_1, \ldots, \mathbf{h}_N]^\top \in \mathbb{R}^{N \times d}$ using the channel-value mixing strategy described in Section 3.6, augmented with standard learnable absolute positional embeddings. These tokens are processed by a Transformer architecture where the self-attention mechanism is modulated by our proposed **Symplectic Positional Embedding (SyPE)**, which rotates queries and keys based on an **adaptive warp module** $\widehat{\tau}$.

**Proof Strategy.** All formal statements and derivations are presented for a single self-attention head within a single Transformer layer, with dimension $d_h$. Since multi-head attention operates by concatenating independent head outputs and deep Transformers are functional compositions of these layers, the properties derived here generalize structurally to the full architecture.

### 3.2. Data Structure: Temporally Warped Seasonal Dynamics

Real-world systems often exhibit *time warping*, where periodic structures persist but the "speed" of physical time flows non-uniformly relative to the sampling index. We formalize this via a monotone warping function $\tau : \{1, \ldots, N\} \to \mathbb{R}_+$. Consider a warped seasonal AR(1) process for a scalar sequence $x_1, \ldots, x_N$:

$$x_t = \phi x_{t-1} + A \sin\left(\omega_0 \tau(t)\right) + \epsilon_t, \quad (1)$$

where $\epsilon_t \sim \mathcal{N}(0, \sigma^2)$. In this formulation, correlations do not peak at a fixed index lag $\delta = |m - n|$, but rather when the *warped time difference* $\Delta\tau_{m,n} = |\tau(m) - \tau(n)|$ aligns with the underlying period $2\pi k/\omega_0$.

### 3.3. Single-Layer Self-Attention with Position Modulation

We define the attention computation on the sequence of hidden representations $\mathbf{H} \in \mathbb{R}^{N \times d}$. For a single head, let $\mathbf{W}_Q, \mathbf{W}_K, \mathbf{W}_V \in \mathbb{R}^{d_h \times d}$ be the projection matrices. The query, key, and value vectors at step $t$ are:

$$\mathbf{q}_t = \mathbf{W}_Q \mathbf{h}_t, \quad \mathbf{k}_t = \mathbf{W}_K \mathbf{h}_t, \quad \mathbf{v}_t = \mathbf{W}_V \mathbf{h}_t. \quad (2)$$

Let $t \mapsto \widehat{\tau}_t$ be an adaptive warp module function. We apply a Rotational Position Modulation via a block-diagonal matrix $\mathbf{R}(\cdot)$. This formulation adopts the structural design of Rotary Positional Embeddings (RoPE) (Su et al., 2024); specifically, if $\mathbf{R}$ is restricted to standard fixed-frequency rotations and the time step is static ($\widehat{\tau}_t = t$), our formulation recovers standard RoPE exactly. In the general case, the modulated queries and keys are:

$$\tilde{\mathbf{q}}_m = \mathbf{R}(\widehat{\tau}_m)\mathbf{q}_m, \quad \tilde{\mathbf{k}}_n = \mathbf{R}(\widehat{\tau}_n)\mathbf{k}_n. \quad (3)$$

The resulting attention score $s_{m,n}$ exploits the orthogonality of $\mathbf{R}$ to depend only on the relative warped time displace-

ment:

$$s_{m,n} = \frac{(\tilde{\mathbf{q}}_m)^\top \tilde{\mathbf{k}}_n}{\sqrt{d_h}} = \frac{\mathbf{q}_m^\top \mathbf{R}(\hat{\tau}_n - \hat{\tau}_m)\mathbf{k}_n}{\sqrt{d_h}}. \qquad (4)$$

The attention weights $a_{m,n}$ and the head output $\mathbf{o}_m$ follow standard definitions: $a_{m,n} = \text{softmax}(s_{m,n})$ and $\mathbf{o}_m = \sum_{n=1}^{N} a_{m,n}\mathbf{v}_n$.

### 3.4. Impossibility of Standard RoPE

We demonstrate that standard Rotary Positional Embeddings (RoPE) are mathematically incapable of representing warped dynamics when the underlying time warping function $\tau$ is non-affine. This limitation arises because RoPE enforces a stationary angular velocity (constant frequency), whereas non-affine warping implies a time-varying instantaneous frequency. This motivates the need for our adaptive warp module $\hat{\tau}$ introduced in Section 3.5.

**Theorem 3.1** (Impossibility of RoPE for Non-Affine Warping). *Let $\tau : \{1, \ldots, N\} \to \mathbb{R}_+$ be a non-affine function. Assume the non-aliasing condition $|\omega_0(\tau(t+1)-\tau(t))| < \pi$ for all $t$. Then there exists no $\theta \in \mathbb{R}$ satisfying the RoPE relative position property:*

$$\theta(m - n) \equiv \omega_0(\tau(m) - \tau(n)) \pmod{2\pi} \quad \forall m, n.$$

*Proof.* Assume such $\theta$ exists with representative $\theta \in (-\pi, \pi]$. Let $\Delta\tau(t) = \tau(t+1)-\tau(t)$. Setting $(m, n) = (t+1, t)$ implies $\theta \equiv \omega_0\Delta\tau(t) \pmod{2\pi}$. Thus, $\omega_0\Delta\tau(t) = \theta + 2\pi k_t$ for some $k_t \in \mathbb{Z}$. By the non-aliasing condition, $|\omega_0\Delta\tau(t)| < \pi$. However, if $k_t \neq 0$, the reverse triangle inequality yields:

$$|\omega_0\Delta\tau(t)| = |\theta + 2\pi k_t| \geq 2\pi|k_t| - |\theta| \geq 2\pi - \pi = \pi,$$

a contradiction. Hence $k_t = 0$ for all $t$, implying $\Delta\tau(t) = \theta/\omega_0$ is constant. A function with constant increments is affine, contradicting the hypothesis. $\square$

**Remark.** This impossibility result holds strictly for *non-affine* temporal warping under the stated non-aliasing condition. Under purely affine warping (e.g., $\tau(t) = at + b$), RoPE trivially absorbs the constant scaling into its static frequency. Furthermore, the non-aliasing constraint is required to prevent pathological phase-wrapping from artifactually satisfying the congruence.

For further analysis on other positional embedding options like additive position encoding, refer to appendix C

### 3.5. Method: Symplectic Positional Embeddings (SyPE)

To resolve the limitation in Theorem 3.1, we introduce SyPE, which replaces static RoPE rotations with a dynamic symplectic flow $\mathbf{S}(\Delta\hat{\tau})$ driven by an adaptive warp module.

**Symplectic Flow Formulation.** We define the position encoding in the symplectic group $\text{Sp}(2, \mathbb{R})$. For each frequency band, we parameterize a symmetric Hamiltonian matrix $\mathbf{K} = \begin{pmatrix} a & c \\ c & b \end{pmatrix}$ with $a, b > 0$ and $ab - c^2 > 0$. This generates a continuous flow $\mathbf{S}(t) = \exp(t\mathbf{J}\mathbf{K})$, where $\mathbf{J} = \begin{pmatrix} 0 & 1 \\ -1 & 0 \end{pmatrix}$ is the standard symplectic matrix.

**Structured Generalization of RoPE.** SyPE provides a structured Hamiltonian parameterization of rotational modulation that includes RoPE as a special case (detailed derivation provided in the Appendix). While this formulation does not introduce an absolute-time gating mechanism by itself, the symplectic geometric prior offers a flexible inductive bias that may empirically improve training dynamics and conditioning by allowing the model to adaptively warp temporal distances beyond the constraints of rigid RoPE rotations.

**Adaptive Warp Module.** The warped time $\hat{\tau}$ is computed dynamically from the input $\mathbf{X}$. Let $\mathbf{h}_t$ be the representation at step $t$. We compute local time increments:

$$\Delta\hat{\tau}_t = \text{Softplus}(\mathbf{w}_\tau^\top \mathbf{h}_t), \quad \hat{\tau}_m = \sum_{i=1}^{m} \Delta\hat{\tau}_i. \qquad (5)$$

**SyPE Attention Mechanism.** We apply the symplectic flow to the query and its conjugate to the key. For subvectors $\mathbf{q}, \mathbf{k} \in \mathbb{R}^2$:

$$\tilde{\mathbf{q}}_m = \mathbf{S}(\hat{\tau}_m)\mathbf{q}_m, \quad \tilde{\mathbf{k}}_n = \mathbf{J}\mathbf{S}(\hat{\tau}_n)\mathbf{k}_n. \qquad (6)$$

**Theorem 3.2** (SyPE Representations of Warped Time). *The SyPE attention score depends exclusively on the warped time difference:*

$$\langle \tilde{\mathbf{q}}_m, \tilde{\mathbf{k}}_n \rangle = \mathbf{q}_m^\top \mathbf{J}\mathbf{S}(\hat{\tau}_n - \hat{\tau}_m)\mathbf{k}_n. \qquad (7)$$

*Proof.* Using the property $\mathbf{S}(t)^\top \mathbf{J}\mathbf{S}(t) = \mathbf{J}$ and the group property $\mathbf{S}(t)\mathbf{S}(u) = \mathbf{S}(t + u)$:

$$\langle \tilde{\mathbf{q}}_m, \tilde{\mathbf{k}}_n \rangle = (\mathbf{S}(\hat{\tau}_m)\mathbf{q}_m)^\top \mathbf{J}\mathbf{S}(\hat{\tau}_n)\mathbf{k}_n \qquad (8)$$

$$= \mathbf{q}_m^\top \mathbf{S}(\hat{\tau}_m)^\top \mathbf{J}\mathbf{S}(\hat{\tau}_n)\mathbf{k}_n \qquad (9)$$

$$= \mathbf{q}_m^\top \left[\mathbf{S}(\hat{\tau}_m)^\top \mathbf{J}\mathbf{S}(\hat{\tau}_m)\right] \mathbf{S}(-\hat{\tau}_m)\mathbf{S}(\hat{\tau}_n)\mathbf{k}_n \qquad (10)$$

$$= \mathbf{q}_m^\top \mathbf{J}\mathbf{S}(\hat{\tau}_n - \hat{\tau}_m)\mathbf{k}_n. \qquad (11)$$

$\square$

### 3.6. StretchTime: Multivariate Time Series Forecasting with SyPE

**Last Value Residual Learning.** To mitigate the impact of non-stationarity and distribution shifts, we employ a residual learning strategy anchored to the last observed value.

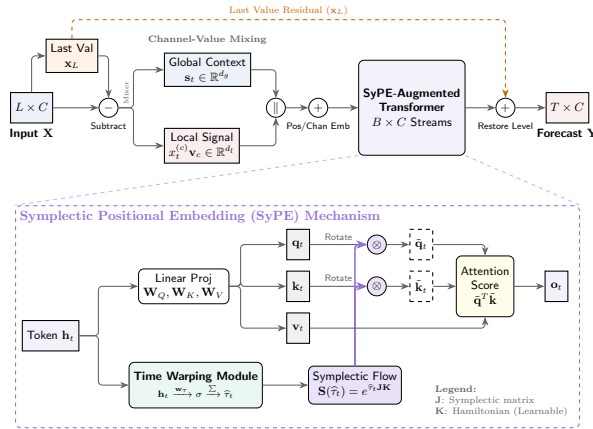

*Figure 2.* Overview of the SyPE-Augmented Transformer architecture.

Unlike statistical normalization methods (e.g., RevIN or Z-score) that normalize based on window statistics, we strictly center the input sequence $\mathbf{X}$ relative to the final time step of the lookback window, denoted as $\mathbf{x}_L$. The model processes the relative difference sequence $\mathbf{X}_{\text{diff}} = \mathbf{X} - \mathbf{x}_L$, effectively learning to forecast the temporal increments rather than absolute magnitudes. The final prediction $\widehat{\mathbf{Y}}$ is reconstructed via a global residual connection that restores the reference value:

$$\widehat{\mathbf{Y}} = \mathcal{M}(\mathbf{X}_{\text{diff}}) + \mathbf{x}_L, \tag{12}$$

where $\mathcal{M}$ denotes the forecasting model. This approach preserves the global level of the series while allowing the network to focus on modeling local temporal dynamics.

**Channel–Value Mixed Tokenization.** Our tokenization strategy is designed to capture both global multivariate context and local univariate specifics. For a target channel $c$ at time step $t$, we construct a composite token $\mathbf{z}_t^{(c)}$ via concatenation:

$$\mathbf{z}_t^{(c)} = [\mathbf{s}_t \, ; \, x_t^{(c)} \mathbf{v}_c] + \mathbf{p}_t + \mathbf{e}_c, \tag{13}$$

where:

- $\mathbf{s}_t \in \mathbb{R}^{d_{\text{global}}}$ is a *global context vector* derived from a linear projection of all input channels at step $t$.

- $x_t^{(c)} \mathbf{v}_c \in \mathbb{R}^{d_{\text{local}}}$ represents the *local signal*, computed by projecting the scalar value $x_t^{(c)}$ using a channel-specific basis $\mathbf{v}_c$.

- $\mathbf{p}_t$ is a shared learnable absolute positional embedding, and $\mathbf{e}_c$ is a learnable channel identifier.

This design allows the model to attend to global correlations ($\mathbf{s}_t$) while preserving the distinct identity and magnitude of individual series.

**Heterogeneous Channel-Specific Warping.** Crucially, StretchTime does not assume a single, universally shared temporal warp across all channels. Because our architecture processes the sequence via the Channel–Value Mixed tokenization described above, the resulting composite tokens $\mathbf{z}_t^{(c)}$, and their subsequent hidden representations $\mathbf{h}_t^{(c)}$, are independently formulated for each channel $c$. Consequently, the adaptive warp module computes temporal increments that are inherently channel-dependent:

$$\Delta \widehat{\tau}_t^{(c)} = \text{Softplus}(\mathbf{w}_\tau^\top \mathbf{h}_t^{(c)}). \tag{14}$$

This formulation allows different variables within the same multivariate system to undergo entirely distinct temporal deformations. Furthermore, within our multi-head attention framework, each head is parameterized by a distinct symplectic geometry (via a unique Hamiltonian $\mathbf{K}$). The channel-specific queries and keys dynamically determine how to interact with these varying geometries through their bilinear forms, providing the model with immense representational flexibility to capture heterogeneous warping dynamics across different time series variables simultaneously.

**Random Ratio Channel Dropout.** Following the practice in recent time series forecasting advancements (Lu et al., 2025), we implement a random ratio channel dropout strategy to enhance model generalization. During training, we randomly zero out a subset of multivariate channels using a sample-specific keep ratio. The surviving channels are rescaled by the inverse of the keep ratio to preserve the aggregate signal magnitude. This mitigates channel-specific noise and encourages robust, channel-invariant representations.

## 4. Experiments

### 4.1. Synthetic Tasks: Recovering Warped Dynamics

To strictly validate the theoretical motivation behind StretchTime, we evaluate the model's ability to recover signal dynamics under controlled temporal non-stationarity, isolating the positional embedding's contribution from confounding real-world noise.

**Data Generation Process.** We generate synthetic datasets using the Temporally Warped Seasonal AR(1) process defined in Section 3.2 (Eq. 1). The underlying time warping function $\tau(t)$ is instantiated as a non-affine oscillating flow, $\tau(t) = \sum_{i=0}^{t} (1 + A \sin(2\pi i / P))$, simulating a system undergoing periodic acceleration and deceleration. To mirror the scale and dimensionality of standard long-term forecasting benchmarks, we align the dataset configuration with ETTh1, generating a multivariate time series ($C = 7$) of length $N = 17,420$ with hourly granularity. We use a lookback window of $L = 96$ and evaluate on forecasting horizons $T \in \{96, 192, 336, 720\}$.

*Table 1.* Results on the synthetic dataset with temporally warped seasonal dynamics (see Section 3.2). We report MSE and MAE for varying prediction lengths ($H$). Best results are in **bold** and second best are underlined.

| Horizon | StretchTime MSE | StretchTime MAE | RoPE MSE | RoPE MAE | w/o MLP MSE | w/o MLP MAE |
|---|---|---|---|---|---|---|
| 96 | **0.053** | **0.180** | 0.058 | 0.186 | 0.078 | 0.218 |
| 192 | **0.073** | **0.207** | 0.084 | 0.219 | 0.105 | 0.250 |
| 336 | **0.120** | **0.255** | 0.184 | 0.311 | 0.161 | 0.298 |
| 720 | **0.331** | **0.416** | 0.411 | 0.480 | 0.358 | 0.439 |
| Avg | **0.144** | **0.265** | 0.184 | 0.299 | 0.176 | 0.301 |

**Model Configuration and Baselines.** We compare StretchTime against a standard Transformer equipped with traditional RoPE, which we will denote as RoPE. As derived in Theorem 3.1, standard RoPE enforces a stationary angular velocity and is theoretically ill-suited for the accelerating dynamics of our warped dataset. We also include an ablation without MLPs, removing the channel-mixing components, to isolate the contribution of the symplectic geometry driven by the adaptive warp module, which we will denote as w/o MLP. To strictly align the empirical setup with our theoretical derivations assuming a single self-attention layer, we configure all models with a depth of $N = 1$. For more hyperparameter and implementation details, see D.2.

**Results and Analysis.** The quantitative results are summarized in Table 1. StretchTime achieves the lowest MSE and MAE across all prediction horizons, demonstrating robust generalization. Crucially, the performance gap widens significantly as the horizon extends (e.g., a **19.5%** reduction in MSE compared to RoPE at $T = 720$). Most notably, the w/o MLP variant surpasses the full RoPE baseline at longer horizons ($T = 336$ and $T = 720$). This result is pivotal: it confirms that in regimes of temporal warping, the ability to manipulate the temporal dimension via a learnable clock is more valuable than the universal approximation capacity of standard MLPs.

To further validate these findings, we visualize forecast dynamics across multiple test samples in Figure 3. The qualitative comparison reveals a distinct failure mode in the RoPE baseline (right column). While RoPE generally captures the amplitude envelope of the signal, it suffers from severe *phase decoherence* in the prediction window. Because standard RoPE enforces a static rotational velocity, it cannot adapt to the variable frequency of the warped signal, causing the predicted peaks and troughs (blue) to drift out of sync with the ground truth (orange). This desynchronization is systematic across samples. In contrast, StretchTime (left column) maintains tight phase alignment throughout the sequence. The model successfully tracks the shifting periodicity, empirically proving that the symplectic clock $\hat{\tau}$

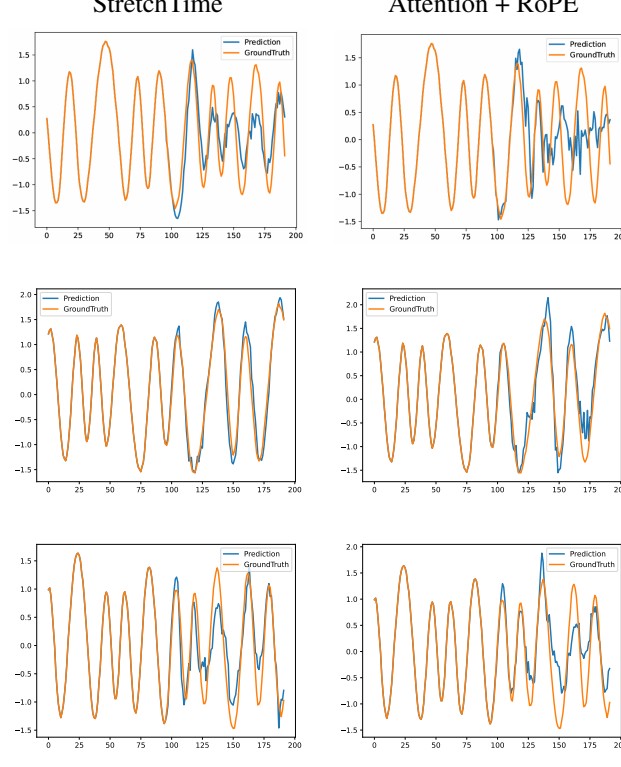

*Figure 3.* Forecast visualization on warped seasonal dynamics. StretchTime (left) corrects the phase alignment errors observed in the static RoPE baseline (right).

effectively "warps" the attention mechanism to match the non-uniform flow of the underlying system.

### 4.2. Multivariate Time Series Forecasting

We conducted comprehensive experiments on 12 widely used multivariate time series forecasting benchmarks, including the four ETT datasets (ETTh1, ETTh2, ETTm1, ETTm2), four PEMS datasets (PEMS03, PEMS04, PEMS08), Weather, Solar-Energy, and Electricity (ECL). These datasets represent diverse domains characterized by complex temporal dependencies and varying channel correlations; detailed descriptions for each are provided in Appendix D.

**Implementation Details** For our proposed StretchTime, we employed a streamlined configuration with $N = 3$ Transformer encoder layers to prevent overfitting. To maintain parameter efficiency across varying channel counts $C$, the hidden dimension was set to $d_{\mathrm{model}} = 64$ for smaller benchmarks and $d_{\mathrm{model}} = 128$ for larger ones. We maintained a consistent effective batch size of 32, employing gradient accumulation on varying physical batch sizes (2–32) to accommodate hardware constraints.

Crucially, to guarantee the fairest possible comparison, we standardized our testing environment, data splitting protocols, and evaluation metrics to precisely match those established in the official OLinear (Yue et al., 2025) and TimeMixer++ (Wang et al., 2025) benchmarks. This strict alignment ensures a rigorously controlled and strictly comparable evaluation framework, which can be explicitly verified in our provided codebase (exp/exp_main.py). For more hyperparameter and implementation details, see Appendix D.2.

**Baselines**    We compared StretchTime against a robust suite of state-of-the-art forecasting models, encompassing both linear architectures and recent Transformer variants. To ensure we are evaluating our model against the absolute peak performance of the most competitive recent architectures, we directly report the results for OLinear and TimeMixer++ exactly as published in their original papers.

For the remaining baselines—TimeMixer (Wang et al., 2024), iTransformer (Liu et al., 2024), PatchTST (Nie et al., 2023), TimesNet (Wu et al., 2023), and DLinear (Zeng et al., 2023)—as well as our internal RoPE baseline, we rigorously retrained and evaluated the models locally within our standardized testing environment to guarantee strict comparability. This methodology allows us to definitively verify whether the superiority of the symplectic formulation, proven in the controlled synthetic setting, translates to complex real-world data against peak baseline configurations.

**Main Results**    Table 2 summarizes the forecasting performance across ten diverse benchmarks. StretchTime achieves the best overall performance, securing the highest average rank of 2.20 and ranking first on 4 out of 10 datasets. It demonstrates particular dominance on datasets characterized by complex, non-stationary periodicity, such as the Traffic (PEMS03, PEMS04) and Solar-Energy datasets, where it outperforms the second-best baselines by clear margins (e.g., reducing MSE on PEMS04 from 0.091 to 0.088). While recent linear models like OLinear and TimeMixer++ show strong performance on specific subsets (e.g., Weather, ECL), StretchTime exhibits superior consistency across diverse domains, never dropping below the top tier of competitive models. This robustness suggests that the symplectic geometric prior offers a flexible inductive bias capable of modeling both highly periodic traffic flows and more stochastic weather patterns. For visualizations with the Multivariate Time Series Forecasting benchmarks, we provide an ipython notebook file in our codebase. (stretchtime.ipynb)

**Comparison with RoPE**    To isolate the contribution of the proposed methodology from the base Transformer architecture, we directly compare StretchTime against the RoPE baseline. The results provide validation of our theoretical premise: StretchTime outperforms RoPE on all 10 datasets, often by substantial margins. For instance, on the Solar dataset—which naturally exhibits varying daylight duration (time warping) across seasons—StretchTime reduces the MSE from 0.214 to 0.195. Similarly, on PEMS03, the error drops from 0.107 to 0.094. This consistent dominance confirms that the performance gains are not merely due to the backbone architecture but are specifically driven by the adaptive warp module $\hat{\tau}$ and the symplectic attention mechanism. It empirically proves that real-world time series contain non-stationary "warped" dynamics that static rotational embeddings (RoPE) fail to capture, necessitating the dynamic geometry introduced by SyPE.

### 4.3. Computational Efficiency Analysis

Table 3 details the computational complexity (FLOPs) and parameter counts on the ETTm1 dataset. StretchTime demonstrates a highly favorable trade-off between model capacity and forecasting accuracy, validating the efficiency of our symplectic formulation.

**Parameter Count.** A key advantage of our approach is the ability to model complex non-stationary dynamics without excessive over-parameterization. Compared to leading channel-independent Transformers like PatchTST and iTransformer, which require approximately 10M–14M parameters, StretchTime achieves state-of-the-art performance with only 494K parameters on average—a reduction of over 95%. This confirms that the symplectic inductive bias effectively captures warped temporal structures that standard architectures must approximate through sheer scale.

**Computational Cost vs. Accuracy.** In terms of execution cost, StretchTime significantly outperforms its closest competitor in accuracy, TimeMixer++. While TimeMixer++ incurs a heavy computational burden, with an average of 6.25G FLOPs, StretchTime reduces this cost by approximately 55% with an average of 2.79G FLOPs while maintaining superior predictive performance. Although linear baselines such as DLinear remain the most computationally frugal, they suffer a distinct performance drop (MSE 0.404 vs. 0.367). StretchTime effectively bridges this gap, delivering the high expressivity of deep Transformers with a computational footprint compatible with resource-constrained deployment.

### 4.4. Factorized Ablation Study

To rigorously isolate the contributions of our proposed temporal mechanism (the adaptive warp module) from the geometric representation (the symplectic flow), we design a comprehensive factorized ablation study. We evaluate performance across five distinct architectural variants on the highly stable PEMS04 and PEMS08 traffic datasets, where complex but structured temporal heterogeneity is prominent.

*Table 2.* Summary of Multivariate TSF Results. Averaged test set MSE are reported. Best results are in **bold** and second best are underlined.

| Model | StretchTime | RoPE | OLinear | TimeMixer++ | TimeMixer | iTransformer | PatchTST | TimesNet | DLinear |
|---|---|---|---|---|---|---|---|---|---|
| Weather | 0.237 | 0.244 | 0.237 | **0.226** | 0.240 | 0.258 | 0.265 | 0.259 | 0.265 |
| Solar | **0.195** | 0.214 | 0.215 | 0.203 | 0.216 | 0.233 | 0.287 | 0.403 | 0.330 |
| ECL | 0.174 | 0.179 | **0.159** | 0.165 | 0.182 | 0.178 | 0.216 | 0.192 | 0.225 |
| ETTh1 | 0.424 | 0.449 | 0.424 | **0.419** | 0.447 | 0.454 | 0.516 | 0.458 | 0.461 |
| ETTh2 | 0.384 | 0.401 | 0.367 | **0.339** | 0.364 | 0.383 | 0.391 | 0.414 | 0.563 |
| ETTm1 | **0.367** | 0.379 | 0.374 | 0.369 | 0.381 | 0.407 | 0.406 | 0.400 | 0.404 |
| ETTm2 | 0.276 | 0.287 | 0.270 | **0.269** | 0.275 | 0.288 | 0.290 | 0.291 | 0.354 |
| PEMS03 | **0.094** | 0.107 | 0.095 | 0.165 | 0.167 | 0.113 | 0.180 | 0.147 | 0.278 |
| PEMS04 | **0.088** | 0.090 | 0.091 | 0.136 | 0.185 | 0.111 | 0.195 | 0.129 | 0.295 |
| PEMS08 | 0.118 | 0.125 | **0.113** | 0.200 | 0.226 | 0.150 | 0.280 | 0.193 | 0.379 |
| AvgRank | **2.20** | 4.20 | 2.30 | 2.80 | 5.00 | 5.30 | 7.75 | 6.70 | 8.55 |
| #Top1 | **4** | 0 | 2 | **4** | 0 | 0 | 0 | 0 | 0 |

*Table 3.* Computational efficiency comparison (FLOPs and Parameters) with Average Rank. Models are ordered to match the main results. Best results are in **bold** and second best are underlined.

| Horizon | StretchTime | | TimeMixer++ | | TimeMixer | | iTransformer | | PatchTST | | TimesNet | | DLinear | |
|---|---|---|---|---|---|---|---|---|---|---|---|---|---|---|
| | FLOPs | Params | FLOPs | Params | FLOPs | Params | FLOPs | Params | FLOPs | Params | FLOPs | Params | FLOPs | Params |
| 96 | 1.24G | 471K | 6.24G | 1.19M | 2.77G | 1.13M | 210M | 9.56M | 2.51G | 10.1M | 2.26G | 1.19M | **259K** | **18.6K** |
| 192 | 1.86G | 480K | 6.25G | 1.2M | 2.81G | 1.14M | 211M | 9.61M | 2.52G | 10.6M | 3.38G | 1.2M | **517K** | **37.2K** |
| 336 | 2.79G | 494K | 6.25G | 1.22M | 2.85G | 1.17M | 213M | 9.68M | 2.54G | 11.5M | 5.06G | 1.22M | **905K** | **65.2K** |
| 720 | 5.27G | 530K | 6.26G | 1.27M | 2.98G | 1.24M | 217M | 9.88M | 2.59G | 13.9M | 9.57G | 1.25M | **1.94M** | **140K** |
| Avg Cost | 2.79G | 493.75K | 6.25G | 1.22M | 2.85G | 1.17M | 212.75M | 9.68M | 2.54G | 11.53M | 5.07G | 1.22M | 905.25K | 65.25K |
| AvgRank | 3.75 | 2.00 | 6.75 | 4.25 | 5.00 | 3.00 | 2.00 | 6.00 | 3.75 | 7.00 | 5.75 | 4.75 | **1.00** | **1.00** |

The variants are defined as follows:

- **V1 (Warp Only):** Utilizes the adaptive warp module but ablates positional geometry entirely, relying only on standard attention.

- **V2 (Fixed RoPE):** The standard Transformer baseline with static Rotary Positional Embeddings (Su et al., 2024) and no temporal warping.

- **V3 (Learnable Symplectic):** Replaces RoPE with static-time SyPE ($\widehat{\tau}_t = t$). This tests the capacity of the symplectic geometry independently of temporal warping.

- **V4 (Fixed RoPE + Warp):** Combines the adaptive warp module with standard RoPE. This restricts the continuous flow to the orthogonal subgroup $\mathrm{SO}(2)$.

- **V5 (StretchTime / Full):** Our complete formulation, combining the adaptive warp module with the learnable symplectic geometry $\mathrm{Sp}(2, \mathbb{R})$.

**The Necessity of Symplectic Generalization.** A theoretical examination of the attention mechanism reveals why isolating the geometry to RoPE is fundamentally limiting. Suppose the relative query-key interaction is implemented by a continuous per-token linear flow, such that the resulting bilinear score depends solely on the warped-time difference $\widehat{\tau}_n - \widehat{\tau}_m$. For this to hold across all queries and keys, the flow must strictly preserve the underlying bilinear form, requiring $\mathbf{S}(t)^\top \mathbf{J} \mathbf{S}(t) = \mathbf{J}$. Consequently, the flow must reside in the symplectic group, $\mathbf{S}(t) \in \mathrm{Sp}(2, \mathbb{R})$. Under continuity, its generator is Hamiltonian, $\mathbf{A} = \mathbf{J} \mathbf{K}$ for a symmetric matrix $\mathbf{K}$.

Within this formalization, symplectic geometry is not merely sufficient; it is the exact, requisite geometry class for valid continuous relative position modulation. RoPE emerges simply as an isotropic special case ($\mathbf{K} = \omega \mathbf{I}$). Therefore, replacing SyPE with RoPE (V5 to V4) strictly truncates the realizable warped-time geometry. While such a restriction may sporadically serve as a regularizer on stochastically dominant benchmarks, it fundamentally bottlenecks representational capacity. When temporal heterogeneity is prominent—as in the PEMS traffic networks—the full learnable symplectic generalization is necessary to capture complex state-space evolutions.

**Synergy of Warp and Geometry.** The empirical results in Table 4 corroborate this theoretical standpoint. Neither the adaptive warp module alone (V1) nor the geometry alone (V2, V3) consistently achieves optimal performance. However, extending the rigid geometry of fixed RoPE to the

*Table 4.* Factorized ablation study on stable traffic forecasting benchmarks (MSE). We decouple the adaptive warp module from the underlying positional geometry. Best results are in **bold** and second best are underlined.

| Dataset (Len) | V1
Warp Only | V2
Fixed RoPE | V3
Learnable Symplectic | V4
Fixed RoPE + Warp | V5
Full StretchTime |
|---|---|---|---|---|---|
| PEMS04 (12) | 0.075 | 0.072 | 0.072 | 0.073 | **0.071** |
| PEMS04 (24) | 0.082 | 0.080 | **0.078** | 0.079 | 0.081 |
| PEMS04 (48) | 0.097 | **0.093** | 0.094 | 0.095 | **0.090** |
| PEMS04 (96) | 0.119 | 0.118 | 0.118 | 0.115 | **0.112** |
| **PEMS04 Avg** | 0.093 | 0.091 | 0.091 | 0.090 | **0.089** |
| PEMS08 (12) | 0.075 | 0.075 | 0.076 | 0.076 | **0.070** |
| PEMS08 (24) | 0.097 | 0.105 | 0.105 | 0.101 | **0.099** |
| PEMS08 (48) | 0.132 | 0.133 | 0.130 | **0.129** | 0.122 |
| PEMS08 (96) | 0.198 | 0.201 | **0.188** | 0.195 | 0.182 |
| **PEMS08 Avg** | 0.126 | 0.129 | 0.125 | 0.125 | **0.118** |

learnable symplectic family yields consistent gains across distinct regimes: it improves performance in the absence of temporal warping (**V3 vs. V2**) and further compounds accuracy when coupled with the adaptive warp module (**V5 vs. V4**). These comparisons offer definitive evidence that the symplectic generalization is mathematically principled, not redundant, and empirically indispensable for high-fidelity forecasting.

## 5. Conclusion

This work bridges the gap between static positional encodings and the non-stationary, time-warped dynamics inherent in real-world time series. We demonstrate that standard rotational embeddings (RoPE) are mathematically incapable of representing non-affine temporal progression, leading to phase decoherence in systems with variable periodicity. By generalizing the rotation group to the symplectic group $\mathrm{Sp}(2, \mathbb{R})$ and modulating it via an adaptive warp module, we propose StretchTime, a Transformer architecture that learns to dynamically dilate or contract temporal coordinates. StretchTime achieves state-of-the-art accuracy and superior parameter efficiency compared to existing baselines across diverse benchmarks. As for limitations, we have not yet scaled StretchTime to massive datasets to evaluate its potential as a foundation model for zero-shot forecasting. Additionally, we have not explored applying the symplectic warping framework to general sequence modeling tasks. It would be particularly interesting to investigate whether the richer symplectic representations play a bigger role in complex domains such as natural language processing, potentially offering distinct advantages over standard rotations that are less pronounced in time series forecasting.

## Acknowledgments

This research was supported in part through research cyber-infrastructure resources and services provided by the Partnership for an Advanced Computing Environment (PACE, 2017) at the Georgia Institute of Technology, Atlanta, Georgia, USA.

We thank Lambda, Inc. for their compute resource help.

## Impact Statement

This paper introduces a novel positional encoding framework and architecture for multivariate time series forecasting. As a fundamental architectural improvement, StretchTime primarily affects upstream sequence modeling capabilities rather than specific end-user applications. The positive impacts include substantial reductions in parameter counts and computational costs, alongside improved forecasting accuracy for real-world systems exhibiting complex, time-warped dynamics—benefiting critical domains such as climate science, healthcare monitoring, and infrastructure planning. Like most foundational ML research, this work could indirectly contribute to both beneficial and potentially harmful applications (e.g., surveillance tracking or financial market manipulation) depending on how downstream models implement it. However, as an architectural component rather than a deployed system, StretchTime itself poses minimal direct societal concerns.

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

# A. Appendix: Background on Symplectic Geometry

While symplectic geometry is a foundational discipline in physics and differential geometry, we recognize it is not standard background in the time series forecasting literature. This section provides a brief, intuitive primer on the core concepts of symplectic geometry, specifically Hamiltonian flows and phase space, and explains why it provides the ideal mathematical framework for modeling time-warped sequential dynamics.

**Phase Space and the Symplectic Form.**  Symplectic geometry is the natural geometry of classical mechanics. In mechanics, a physical system is described by its state in a "phase space," which consists of paired coordinates: position ($q$) and momentum ($p$). A symplectic space is equipped with a non-degenerate, skew-symmetric bilinear form (the symplectic form), typically represented by the matrix $\mathbf{J} = \begin{pmatrix} 0 & 1 \\ -1 & 0 \end{pmatrix}$. This form provides a way to measure 2-dimensional oriented areas within the phase space.

**Hamiltonian Flows and Conservation.**  The temporal evolution of a system in phase space is dictated by a scalar energy function called the Hamiltonian ($H$). The system evolves according to Hamilton's equations, generating what is known as a *Hamiltonian flow*. A profound property of Hamiltonian flows, formalized by Liouville's theorem, is that they strictly preserve the symplectic structure over time. Geometrically, this means that as the system evolves, a volume of phase space may severely stretch in one direction (e.g., accelerating position) and compress in another (e.g., decreasing momentum), but its fundamental geometric area and topological structure are conserved.

**Connection to Time Series and Temporal Warping.**  In the context of attention mechanisms, we project queries and keys into 2D subspaces. Standard Rotary Positional Embeddings (RoPE) model the relative position between these vectors using rigid rotations belonging to the special orthogonal group $\mathrm{SO}(2)$. A rigid rotation is isotropic; it assumes the "speed" of temporal progression is perfectly uniform, like the steady ticking of a mechanical clock.

However, real-world time series often exhibit "elastic" dynamics—periods of rapid acceleration (stretching) or deceleration (compression). By extending our positional encodings from the rotation group $\mathrm{SO}(2)$ to the symplectic group $\mathrm{Sp}(2, \mathbb{R})$, we upgrade our model of time from a rigid circle to a flexible, area-preserving geometry. Symplectic transformations allow the latent query-key representation to adaptively stretch and compress its temporal coordinates while conserving the underlying structural integrity of the bilinear attention form. Therefore, Hamiltonian flows provide a principled, physically grounded mathematical framework to model sequences where the physical timeline warps, yet periodic correlations persist.

# B. Appendix: Mathematical Proofs and Derivations

This appendix provides the algebraic derivations for the Symplectic Positional Embedding (SyPE), including the proof of symplectic conservation, the closed-form matrix exponential for efficient implementation, and the structural reduction to RoPE.

## B.1. Symplectic Positional Embedding (SyPE) Derivations

**Proof of Symplectic Conservation (Lemma for Theorem 3.2).**  The main text relies on the property that the generated flow $\mathbf{S}(t) = \exp(t\mathbf{JK})$ preserves the symplectic form, i.e., $\mathbf{S}(t)^\top \mathbf{J}\mathbf{S}(t) = \mathbf{J}$. We provide the explicit proof here.

*Proof.* Let $\mathbf{M}(t) = \mathbf{S}(t)^\top \mathbf{J}\mathbf{S}(t)$. At $t = 0$, $\mathbf{M}(0) = \mathbf{I}^\top \mathbf{J}\mathbf{I} = \mathbf{J}$. Differentiating $\mathbf{M}(t)$ with respect to $t$:

$$\frac{d}{dt}\mathbf{M}(t) = \dot{\mathbf{S}}^\top \mathbf{J}\mathbf{S} + \mathbf{S}^\top \mathbf{J}\dot{\mathbf{S}} \tag{15}$$

$$= (\mathbf{JKS})^\top \mathbf{J}\mathbf{S} + \mathbf{S}^\top \mathbf{J}(\mathbf{JKS}) \quad \text{(using } \dot{\mathbf{S}} = \mathbf{AS} = \mathbf{JKS}) \tag{16}$$

$$= \mathbf{S}^\top \mathbf{K}^\top \mathbf{J}^\top \mathbf{J}\mathbf{S} + \mathbf{S}^\top \mathbf{J}^2 \mathbf{KS}. \tag{17}$$

Recall the properties of the symplectic matrix $\mathbf{J} = \begin{pmatrix} 0 & 1 \\ -1 & 0 \end{pmatrix}$: $\mathbf{J}^\top = -\mathbf{J}$, $\mathbf{J}^2 = -\mathbf{I}$, and $\mathbf{J}^\top \mathbf{J} = \mathbf{I}$. Also, $\mathbf{K}$ is symmetric

$(\mathbf{K}^\top = \mathbf{K})$. Substituting these:

$$\frac{d}{dt}\mathbf{M}(t) = \mathbf{S}^\top\mathbf{K}(-\mathbf{J})\mathbf{J}\mathbf{S} + \mathbf{S}^\top(-\mathbf{I})\mathbf{K}\mathbf{S} \tag{18}$$

$$= \mathbf{S}^\top\mathbf{K}(-\mathbf{J}^2)\mathbf{S} - \mathbf{S}^\top\mathbf{K}\mathbf{S} \tag{19}$$

$$= \mathbf{S}^\top\mathbf{K}(\mathbf{I})\mathbf{S} - \mathbf{S}^\top\mathbf{K}\mathbf{S} \tag{20}$$

$$= 0. \tag{21}$$

Since the derivative is zero for all $t$, $\mathbf{M}(t)$ is constant and equals $\mathbf{M}(0) = \mathbf{J}$. $\qquad\square$

**Closed-Form Implementation of Symplectic Flow.** To avoid computationally expensive matrix exponentiation during training, we derive the closed-form solution for $\mathbf{S}(t) = \exp(t\mathbf{A})$, where $\mathbf{A} = \mathbf{J}\mathbf{K} = \left(\begin{smallmatrix} c & b \\ -a & -c \end{smallmatrix}\right)$.

Compute the square of the generator $\mathbf{A}$:

$$\mathbf{A}^2 = \begin{pmatrix} c & b \\ -a & -c \end{pmatrix}\begin{pmatrix} c & b \\ -a & -c \end{pmatrix} = \begin{pmatrix} c^2 - ab & bc - bc \\ -ac + ac & -ab + c^2 \end{pmatrix} = -(ab - c^2)\mathbf{I}. \tag{22}$$

Let $\omega^2 = ab - c^2$. Provided the stability constraint $ab - c^2 > 0$ holds (see below), we can define the frequency $\omega = \sqrt{ab - c^2}$. The power series for the matrix exponential splits into even and odd terms:

$$\mathbf{S}(t) = e^{t\mathbf{A}} = \sum_{k=0}^{\infty}\frac{t^{2k}}{(2k)!}\mathbf{A}^{2k} + \sum_{k=0}^{\infty}\frac{t^{2k+1}}{(2k+1)!}\mathbf{A}^{2k+1} \tag{23}$$

$$= \left(\sum_{k=0}^{\infty}\frac{(-1)^k(\omega t)^{2k}}{(2k)!}\right)\mathbf{I} + \left(\sum_{k=0}^{\infty}\frac{(-1)^k(\omega t)^{2k+1}}{(2k+1)!}\right)\frac{\mathbf{A}}{\omega} \tag{24}$$

$$= \cos(\omega t)\mathbf{I} + \frac{\sin(\omega t)}{\omega}\mathbf{A}. \tag{25}$$

This formula allows efficient $\mathcal{O}(1)$ computation of the flow for every token.

**Stability Constraints and Parameterization.** For the flow to be oscillatory (stable) rather than hyperbolic (divergent), the generator $\mathbf{A}$ must have purely imaginary eigenvalues. The characteristic equation is $\lambda^2 + (ab - c^2) = 0$. Stability requires the determinant of $\mathbf{K}$ to be positive: $\det(\mathbf{K}) = ab - c^2 > 0$. We enforce this via the following parameterization during training:

$$a = e^\alpha, \quad b = e^\beta, \quad c = \rho\sqrt{ab} \quad \text{with } \rho = \tanh(\gamma) \in (-1, 1). \tag{26}$$

This guarantees $ab - c^2 = ab(1 - \rho^2) > 0$, ensuring strictly stable dynamics.

## B.2. Relationship to RoPE

**RoPE as a Special Case.** We demonstrate that SyPE strictly generalizes Rotary Positional Embeddings. Consider the case where the Hamiltonian is isotropic, $\mathbf{K} = \omega\mathbf{I}$. The generator becomes $\mathbf{A} = \mathbf{J}(\omega\mathbf{I}) = \omega\mathbf{J}$. The flow becomes:

$$\mathbf{S}(t) = \exp(t\omega\mathbf{J}) = \cos(\omega t)\mathbf{I} + \sin(\omega t)\mathbf{J} = \begin{pmatrix} \cos(\omega t) & \sin(\omega t) \\ -\sin(\omega t) & \cos(\omega t) \end{pmatrix}. \tag{27}$$

This is exactly the rotation matrix $\mathbf{R}(\omega t)$ used in RoPE. Under SyPE, the interaction term is $\mathbf{q}^\top\mathbf{J}\mathbf{S}(\Delta t)\mathbf{k}$. Substituting $\mathbf{S}(\Delta t) = \mathbf{R}(\omega\Delta t)$:

$$\mathbf{q}^\top\mathbf{J}\mathbf{R}(\theta)\mathbf{k}. \tag{28}$$

By choosing orthogonal basis vectors for $\mathbf{q}$ and $\mathbf{k}$ (e.g., $\mathbf{q} = [0, 1]^\top, \mathbf{k} = [1, 0]^\top$), this recovers the standard relative attention score $\cos(\theta)$, confirming that RoPE is the specific instance of SyPE where the Hamiltonian is fixed to the identity matrix.

**Extension to High Dimensions.** Standard RoPE applies 2D rotations to pairs of dimensions. SyPE follows the same structural logic. For a head dimension $d_h$, we decompose the space into $d_h/2$ disjoint 2D subspaces. We learn a separate Hamiltonian $\mathbf{K}_i$ (and thus a separate flow $\mathbf{S}_i$) for each subspace $i$. The full transformation is block-diagonal:

$$\mathbf{S}_{total}(t) = \text{blkdiag}\left(\mathbf{S}_1(t), \mathbf{S}_2(t), \ldots, \mathbf{S}_{d_h/2}(t)\right). \tag{29}$$

This preserves the computational efficiency of RoPE while allowing each frequency band to learn its own anisotropic time-warping dynamics.

# C. Appendix: Limitations of Other Positional Encodings

In the main text, we demonstrated that Rotary Positional Embeddings (RoPE) cannot represent non-affine time warping because they enforce a stationary rotational frequency. Here, we extend this analysis to other common encoding schemes: Relative Positional Encodings (RPE) and Learnable Additive/Absolute Positional Encodings (APE).

## C.1. Relative Positional Encodings (RPE)

Standard RPE methods (e.g., T5, ALiBi) inject a bias term $B_{m,n}$ into the attention scores that depends solely on the index distance $m - n$. We show that this translation-invariant formulation is fundamentally incompatible with time warping, where the "physical" duration of a step varies over time.

**Proposition C.1** (Inconsistency of RPE with Warped Time). *Let $\tau : \mathbb{N} \to \mathbb{R}_+$ be the underlying warping function. An RPE mechanism defines a bias $b : \mathbb{Z} \to \mathbb{R}$ such that the attention modification is $B_{m,n} = b(m - n)$. If $\tau$ is non-affine, there exists no function $b(\cdot)$ such that the attention bias depends solely on the warped time distance $\Delta\tau_{m,n} = \tau(m) - \tau(n)$.*

*Proof.* We aim to find a bias function $b$ such that $b(m - n) = g(\tau(m) - \tau(n))$ for some injective function $g$ (representing the desired dependency on warped time).

Consider a fixed index lag $k \in \mathbb{Z} \setminus \{0\}$. The RPE bias $b(k)$ is a constant value for all pairs $(m, n)$ where $m - n = k$. However, the warped time difference for this fixed lag is $\delta_k(n) = \tau(n + k) - \tau(n)$. If $\tau$ is non-affine, the discrete derivative (and by extension the $k$-step difference) is not constant; $\delta_k(n)$ varies with $n$. Thus, we require a constant $b(k)$ to map to a variable $\delta_k(n)$ via $g$, which is a contradiction. The RPE mechanism will assign the exact same positional bias to pairs $(n + k, n)$ regardless of whether the physical time elapsed between them is short (compressed time) or long (stretched time). $\square$

*Remark* C.2 (Intuition). RPE assumes that "5 steps ago" always means the same thing. In a warped system (e.g., a process accelerating), "5 steps ago" might represent 1 second of physical time early in the sequence, but 0.1 seconds later on. RPE cannot distinguish these cases.

## C.2. Additive (Absolute) Positional Encodings (APE)

Learnable Additive PE assigns a unique vector $\mathbf{p}_t \in \mathbb{R}^d$ to each time step $t$, which is added to the input: $\mathbf{z}_t = \mathbf{x}_t + \mathbf{p}_t$. While a learnable APE is highly flexible and can theoretically memorize the warping of a *single* univariate series (by overfitting $\mathbf{p}_t$ to the specific value required at step $t$), it fails in the **Multivariate Channel-Independent (CI)** setting.

In modern CI forecasting (e.g., PatchTST, iTransformer), the same model parameters (and usually the same positional embeddings) are shared across all channels to enable generalization. We show that a shared APE cannot model a system where different channels undergo distinct warping dynamics (heterogeneity).

**Proposition C.3** (Impossibility of Shared APE for Heterogeneous Warping). *Consider a multivariate system with $C$ channels. Let the dynamics of channel $i$ be governed by a function $f$ of its specific warped clock $\tau_i(t)$:*

$$x_t^{(i)} = f(\tau_i(t))$$

*Assume a Channel-Independent Transformer where a shared positional embedding $\mathbf{p}_t$ is added to all channels, and the network learns a shared mapping $\mathcal{F} : \mathbb{R}^d \to \mathbb{R}$ to predict values based on position. If there exist two channels $i, j$ with distinct warping functions $\tau_i \neq \tau_j$ (and $f$ is non-trivial), there exists no single sequence of embeddings $\{\mathbf{p}_t\}_{t=1}^N$ that allows $\mathcal{F}$ to correctly predict both signals simultaneously.*

*Proof.* For the network to predict the signal value $x_t^{(c)}$ solely from the position (a simplified proxy for time-dependent dynamics), it must learn a mapping $\mathcal{F}$ such that $\mathcal{F}(\mathbf{p}_t) \approx f(\tau_c(t))$. This implies:

$$\mathbf{p}_t \approx \mathcal{F}^{-1}(f(\tau_c(t))).$$

For this to hold for channel $i$, we require $\mathbf{p}_t$ to encode $\tau_i(t)$. For this to hold for channel $j$, we require $\mathbf{p}_t$ to encode $\tau_j(t)$. If $\tau_i(t) \neq \tau_j(t)$ (e.g., channel $i$ is accelerating while channel $j$ is constant), we have a conflict: $\mathbf{p}_t$ must simultaneously map to two different values in the domain of $f$. Since $\mathbf{p}_t$ is a fixed vector for index $t$, it cannot satisfy these contradictory constraints. $\square$

*Remark* C.4 (The "One-Size-Fits-None" Problem). In a multivariate dataset, Channel A might exhibit a "fast" seasonal cycle (e.g., weekly) while Channel B exhibits a "slow" cycle (e.g., monthly), or they may speed up/slow down at different rates. A standard Additive PE forces the model to learn a single "average" notion of time for index $t$. Consequently, the model will likely underfit both channels, failing to capture the sharp peaks of the fast channel and the broad plateaus of the slow channel. Our method (SyPE) resolves this by dynamically stretching the clock $\hat{\tau}$ specifically for each channel/sample.

# D. Appendix: Datasets and Experimental Details

## D.1. Experimental Datasets

We evaluate our model on eight widely used multivariate time series benchmark datasets. Table 2 summarizes the statistics of these datasets. The full names and important features of these datasets are summarized as follows:

**Weather Dataset (Wu et al., 2021):** This dataset contains local climatological data recorded every 10 minutes for 2020, including 21 meteorological indicators such as air temperature, humidity, and wind speed.

**Solar Dataset (Lai et al., 2018):** The Solar dataset contains solar power production records from 137 photovoltaic plants in Alabama State in 2006, sampled every 10 minutes.

**Electricity Dataset (Wu et al., 2021):** This dataset records the hourly electricity consumption of 321 consumers. It captures long-term dependencies and periodic patterns in energy usage over a three-year period from 2012 to 2014.

**ETT Dataset (Zhou et al., 2021):** The ETT (Electricity Transformer Temperature) dataset consists of data collected from electricity transformers, including load and oil temperature, recorded every 15 minutes (ETTm1, ETTm2) and every hour (ETTh1, ETTh2) over a period of two years (2016-2018).

**PEMS Dataset (Li et al., 2018):** This dataset consists of public traffic network data collected by the California Transportation Agencies (CalTrans) Performance Measurement System (PeMS). The data is sampled at 5-minute intervals from sensors deployed across various districts in California to capture complex spatial-temporal dependencies. In this work, we utilize three widely adopted subsets, specifically PEMS03, PEMS04, and PEMS08, which serve as established benchmarks for evaluating traffic flow forecasting across networks ranging from 170 to over 800 sensors.

**Additional Discussion on Baselines**

**Computational Constraints:** We exclude results for the Traffic and PEMS07 datasets due to excessive memory usage on TimeMixer (Wang et al., 2024) with our hardware (NVIDIA RTX4090)

**Baseline Result Sourcing:** We source results for the TimeMixer++ (Wang et al., 2025) and OLinear (Yue et al., 2025) baselines directly from their original publications. For TimeMixer++, we rely on the reported figures as we were unable to reproduce the performance using the official codebase under our experimental settings.

**Exchange Dataset Exclusion:** We omit the Exchange dataset as financial time series are dominated by random walk properties (Fama, 1970). As demonstrated by PatchTST (Nie et al., 2023), trivial "last-value" baselines perform comparably to state-of-the-art models on this benchmark, indicating that it provides limited value for evaluating learned long-term temporal dependencies.

## D.2. Hyper-parameter Settings and Implementation Details

**Model Architecture - Multivariate Time Series Forecasting.** For the hyper-parameter settings of StretchTime for multivariate time series forecasting tasks, we use $N = 3$ Transformer encoder layers and $n_{\text{heads}} = 4$ attention heads. We

adjust the hidden dimension $d_{\text{model}}$ according to dataset scale, utilizing 64 for smaller benchmarks and 128 for larger ones. We utilize a standard feedforward dimension of $4d_{\text{model}}$. A dropout rate of 0.1 was applied only to the *ETT* and *Weather* datasets. We employ the GPT-2 weight initialization scheme. For the symplectic flow parameters, we enforce stability constraints to ensure non-divergent dynamics. We experimented with both LayerNorm and RMSNorm, opting for standard LayerNorm applied before the symplectic attention mechanism.

**Model Architecture - Synthetic Tasks.** For the Synthetic dataset, we follow a similar setup as the multivariate time series forecasting tasks, only changing a few hyperparameters. For consistency with our claims in section 3, we set the number of attention layers $n_{\text{layers}} = 1$, the model dimension $d_{\text{model}} = 128$, feed-forward dimension to $d_{\text{ff}} = 512$, and use $n_{\text{heads}} = 4$ attentions heads. Training is performed using the AdamW optimizer with a learning rate of $2 \times 10^{-4}$ and a dropout rate of 0.1.

**Data Processing Strategy.** To handle non-stationarity, we employ a "Last Value Residual" strategy where the input sequence is centered relative to the last observed value $x_L$, and the mean is added back to the final prediction. All input series are processed using a Channel-Independent strategy where applicable.

**Training and Hardware.** The random seed used in all experiments is 2026. All training tasks in this paper were conducted using a single NVIDIA RTX4090 GPU or L40S GPU. We standardize the effective batch size at 32. To accommodate the larger datasets (Electricity, Solar, PEMS), we use a physical batch size of 2-4 with 8-16 gradient accumulation steps to maintain this effective size. During training, StretchTime is trained using the MSE loss function. We use the AdamW optimizer with a learning rate of $5 \times 10^{-4}$ and a cosine annealing scheduler, with an early-stopping patience set to 12 epochs.

# E. Appendix: Full Multivariate Time Series Forecasting Results Table

*Table 5.* Full Multivariate TSF Results. We report MSE and MAE. Best results are in **bold** and second best are underlined. AvgRank and #Top1 are calculated based on MSE across all horizons.

| Model Dataset | Len | StretchTime MSE | MAE | RoPE MSE | MAE | OLinear MSE | MAE | TimeMixer++ MSE | MAE | TimeMixer MSE | MAE | iTransformer MSE | MAE | PatchTST MSE | MAE | TimesNet MSE | MAE | DLinear MSE | MAE |
|---|---|---|---|---|---|---|---|---|---|---|---|---|---|---|---|---|---|---|---|
| Weather | 96 | **0.143** | 0.203 | 0.146 | 0.206 | 0.153 | **0.190** | 0.155 | 0.205 | 0.163 | 0.209 | 0.174 | 0.214 | 0.186 | 0.227 | 0.172 | 0.220 | 0.195 | 0.252 |
| | 192 | **0.196** | 0.258 | 0.201 | 0.262 | 0.200 | **0.235** | 0.201 | 0.245 | 0.208 | 0.250 | 0.221 | 0.254 | 0.234 | 0.265 | 0.219 | 0.261 | 0.237 | 0.295 |
| | 336 | 0.253 | 0.304 | 0.270 | 0.317 | 0.258 | 0.280 | **0.237** | **0.265** | 0.251 | 0.287 | 0.278 | 0.296 | 0.284 | 0.301 | 0.280 | 0.306 | 0.282 | 0.331 |
| | 720 | 0.355 | 0.373 | 0.360 | 0.374 | 0.337 | **0.333** | **0.312** | 0.334 | 0.339 | 0.341 | 0.358 | 0.347 | 0.356 | 0.349 | 0.365 | 0.359 | 0.345 | 0.382 |
| Solar | 96 | **0.158** | 0.241 | 0.190 | 0.268 | 0.179 | **0.191** | 0.171 | 0.231 | 0.189 | 0.259 | 0.203 | 0.237 | 0.265 | 0.323 | 0.373 | 0.358 | 0.290 | 0.378 |
| | 192 | **0.191** | 0.267 | 0.214 | 0.279 | 0.209 | **0.213** | 0.218 | 0.263 | 0.222 | 0.283 | 0.233 | 0.261 | 0.288 | 0.332 | 0.397 | 0.376 | 0.320 | 0.398 |
| | 336 | 0.214 | 0.293 | 0.233 | 0.294 | 0.231 | **0.229** | 0.212 | 0.269 | 0.231 | 0.292 | 0.248 | 0.273 | 0.301 | 0.339 | 0.420 | 0.380 | 0.353 | 0.415 |
| | 720 | 0.217 | 0.272 | 0.218 | 0.286 | 0.241 | **0.236** | 0.212 | 0.270 | 0.223 | 0.285 | 0.249 | 0.275 | 0.295 | 0.336 | 0.420 | 0.381 | 0.357 | 0.413 |
| ECL | 96 | 0.142 | 0.247 | 0.148 | 0.253 | **0.131** | **0.221** | 0.135 | 0.222 | 0.153 | 0.247 | 0.148 | 0.240 | 0.190 | 0.296 | 0.168 | 0.272 | 0.210 | 0.302 |
| | 192 | 0.161 | 0.260 | 0.163 | 0.266 | 0.150 | **0.238** | **0.147** | **0.235** | 0.166 | 0.256 | 0.162 | 0.253 | 0.199 | 0.304 | 0.184 | 0.322 | 0.210 | 0.305 |
| | 336 | 0.176 | 0.283 | 0.182 | 0.288 | 0.165 | **0.254** | **0.164** | **0.245** | 0.185 | 0.277 | 0.178 | 0.269 | 0.217 | 0.319 | 0.198 | 0.300 | 0.223 | 0.319 |
| | 720 | 0.218 | 0.319 | 0.223 | 0.325 | **0.191** | **0.279** | 0.212 | 0.310 | 0.225 | 0.310 | 0.225 | 0.317 | 0.258 | 0.352 | 0.220 | 0.320 | 0.258 | 0.350 |
| ETTh1 | 96 | 0.371 | 0.400 | 0.381 | 0.407 | **0.360** | **0.382** | 0.361 | 0.403 | 0.375 | 0.400 | 0.386 | 0.405 | 0.460 | 0.447 | 0.384 | 0.402 | 0.397 | 0.412 |
| | 192 | 0.418 | 0.429 | 0.430 | 0.439 | **0.416** | **0.414** | 0.416 | 0.441 | 0.429 | 0.421 | 0.441 | 0.512 | 0.477 | 0.429 | 0.436 | 0.429 | 0.446 | 0.441 |
| | 336 | 0.451 | 0.446 | 0.472 | 0.464 | 0.457 | 0.438 | **0.430** | **0.434** | 0.484 | 0.458 | 0.487 | 0.458 | 0.546 | 0.496 | 0.491 | 0.469 | 0.489 | 0.467 |
| | 720 | **0.455** | 0.470 | 0.511 | 0.512 | 0.463 | 0.462 | 0.467 | **0.451** | 0.498 | 0.482 | 0.503 | 0.491 | 0.544 | 0.517 | 0.521 | 0.500 | 0.513 | 0.510 |
| ETTh2 | 96 | 0.287 | 0.343 | 0.294 | 0.347 | 0.284 | 0.329 | **0.276** | **0.328** | 0.289 | 0.341 | 0.297 | 0.349 | 0.308 | 0.355 | 0.340 | 0.374 | 0.340 | 0.394 |
| | 192 | 0.379 | 0.401 | 0.393 | 0.417 | 0.360 | **0.379** | **0.342** | **0.379** | 0.372 | 0.392 | 0.380 | 0.400 | 0.393 | 0.405 | 0.402 | 0.414 | 0.482 | 0.479 |
| | 336 | 0.428 | 0.447 | 0.470 | 0.480 | 0.409 | 0.415 | **0.346** | **0.398** | 0.386 | 0.414 | 0.428 | 0.432 | 0.427 | 0.436 | 0.452 | 0.452 | 0.591 | 0.541 |
| | 720 | 0.441 | 0.463 | 0.445 | 0.472 | 0.415 | 0.431 | **0.392** | **0.415** | 0.412 | 0.434 | 0.427 | 0.445 | 0.436 | 0.450 | 0.462 | 0.468 | 0.839 | 0.661 |
| ETTm1 | 96 | **0.297** | 0.348 | 0.306 | 0.355 | 0.302 | **0.334** | 0.310 | **0.334** | 0.320 | 0.357 | 0.334 | 0.368 | 0.352 | 0.374 | 0.338 | 0.375 | 0.346 | 0.374 |
| | 192 | 0.349 | 0.382 | 0.357 | 0.387 | 0.357 | 0.363 | **0.348** | **0.362** | 0.361 | 0.381 | 0.390 | 0.393 | 0.374 | 0.387 | 0.374 | 0.387 | 0.382 | 0.391 |
| | 336 | 0.382 | 0.405 | 0.399 | 0.417 | 0.387 | **0.385** | **0.376** | 0.391 | 0.390 | 0.404 | 0.426 | 0.420 | 0.421 | 0.414 | 0.410 | 0.411 | 0.415 | 0.415 |
| | 720 | 0.442 | 0.445 | 0.454 | 0.459 | 0.452 | 0.426 | **0.440** | **0.423** | 0.454 | 0.441 | 0.491 | 0.459 | 0.462 | 0.449 | 0.478 | 0.450 | 0.473 | 0.451 |
| ETTm2 | 96 | **0.168** | 0.254 | 0.172 | 0.257 | 0.169 | 0.249 | 0.170 | **0.245** | 0.175 | 0.258 | 0.180 | 0.264 | 0.183 | 0.270 | 0.187 | 0.267 | 0.193 | 0.293 |
| | 192 | 0.232 | 0.297 | 0.244 | 0.306 | 0.232 | **0.290** | **0.229** | 0.291 | 0.237 | 0.299 | 0.250 | 0.309 | 0.255 | 0.314 | 0.249 | 0.309 | 0.284 | 0.361 |
| | 336 | 0.299 | 0.346 | 0.317 | 0.352 | **0.291** | **0.328** | 0.303 | 0.343 | 0.298 | 0.340 | 0.311 | 0.348 | 0.309 | 0.347 | 0.321 | 0.351 | 0.382 | 0.429 |
| | 720 | 0.405 | 0.402 | 0.416 | 0.417 | 0.389 | **0.387** | **0.373** | 0.399 | 0.391 | 0.396 | 0.412 | 0.407 | 0.412 | 0.404 | 0.408 | 0.403 | 0.558 | 0.525 |
| PEMS03 | 12 | **0.060** | 0.165 | 0.069 | 0.175 | **0.060** | **0.159** | 0.097 | 0.208 | 0.076 | 0.188 | 0.071 | 0.174 | 0.099 | 0.216 | 0.085 | 0.192 | 0.122 | 0.243 |
| | 24 | 0.080 | 0.182 | 0.101 | 0.214 | **0.078** | **0.179** | 0.120 | 0.230 | 0.113 | 0.226 | 0.093 | 0.201 | 0.142 | 0.259 | 0.118 | 0.223 | 0.201 | 0.317 |
| | 48 | **0.098** | 0.211 | 0.112 | 0.223 | 0.104 | **0.210** | 0.170 | 0.272 | 0.191 | 0.292 | 0.125 | 0.236 | 0.211 | 0.319 | 0.155 | 0.260 | 0.333 | 0.425 |
| | 96 | **0.138** | 0.255 | 0.148 | 0.255 | 0.140 | **0.247** | 0.274 | 0.342 | 0.288 | 0.363 | 0.164 | 0.275 | 0.269 | 0.370 | 0.228 | 0.317 | 0.457 | 0.515 |
| PEMS04 | 12 | 0.071 | 0.168 | 0.073 | 0.172 | **0.068** | **0.163** | 0.099 | 0.214 | 0.092 | 0.204 | 0.078 | 0.183 | 0.105 | 0.224 | 0.087 | 0.195 | 0.148 | 0.272 |
| | 24 | 0.081 | 0.184 | **0.079** | 0.185 | 0.079 | **0.176** | 0.115 | 0.231 | 0.128 | 0.243 | 0.095 | 0.205 | 0.153 | 0.275 | 0.103 | 0.215 | 0.224 | 0.340 |
| | 48 | **0.090** | 0.198 | 0.095 | 0.206 | 0.095 | **0.197** | 0.144 | 0.261 | 0.213 | 0.315 | 0.120 | 0.233 | 0.229 | 0.339 | 0.136 | 0.250 | 0.355 | 0.437 |
| | 96 | **0.112** | 0.228 | 0.115 | 0.233 | 0.122 | **0.226** | 0.185 | 0.297 | 0.307 | 0.384 | 0.150 | 0.262 | 0.291 | 0.389 | 0.190 | 0.303 | 0.452 | 0.504 |
| PEMS08 | 12 | 0.070 | 0.170 | 0.076 | 0.177 | **0.068** | **0.159** | 0.119 | 0.222 | 0.091 | 0.201 | 0.079 | 0.182 | 0.168 | 0.232 | 0.112 | 0.212 | 0.154 | 0.276 |
| | 24 | 0.099 | 0.199 | 0.101 | 0.200 | **0.089** | **0.178** | 0.149 | 0.249 | 0.137 | 0.246 | 0.115 | 0.219 | 0.224 | 0.281 | 0.141 | 0.238 | 0.248 | 0.353 |
| | 48 | **0.122** | 0.216 | 0.129 | 0.230 | 0.123 | **0.204** | 0.206 | 0.292 | 0.265 | 0.343 | 0.186 | 0.235 | 0.321 | 0.354 | 0.198 | 0.283 | 0.440 | 0.470 |
| | 96 | 0.182 | 0.243 | 0.195 | 0.270 | **0.173** | **0.236** | 0.329 | 0.355 | 0.410 | 0.407 | 0.221 | 0.267 | 0.408 | 0.417 | 0.320 | 0.351 | 0.674 | 0.565 |
| AvgRank | | 2.27 | | 4.28 | | **2.08** | | 3.17 | | 4.83 | | 5.47 | | 7.42 | | 6.67 | | 8.38 | |
| #Top1 | | 13 | | 1 | | 12 | | **17** | | 0 | | 0 | | 0 | | 0 | | 0 | |

# F. Appendix: Further Ablation Study

*Table 6.* Ablation Study Results. Comparison of StretchTime with component variants. Best results are in **bold** and second best are underlined.

| Model Dataset | Len | StretchTime MSE | MAE | RoPE MSE | MAE | Pure Softmax MSE | MAE | LinAttn+RoPE MSE | MAE |
|---|---|---|---|---|---|---|---|---|---|
| Weather | 96 | **0.143** | **0.203** | 0.146 | 0.206 | 0.146 | 0.209 | 0.152 | 0.211 |
| | 192 | **0.196** | **0.258** | 0.201 | 0.262 | 0.198 | 0.260 | 0.210 | 0.270 |
| | 336 | **0.253** | **0.304** | 0.270 | 0.317 | 0.269 | 0.317 | 0.280 | 0.329 |
| | 720 | **0.355** | **0.373** | 0.360 | 0.374 | 0.359 | 0.378 | 0.369 | 0.378 |
| Solar | 96 | **0.158** | **0.241** | 0.190 | 0.268 | 0.183 | 0.249 | 0.187 | 0.245 |
| | 192 | **0.191** | 0.267 | 0.214 | 0.279 | 0.209 | **0.256** | 0.199 | 0.257 |
| | 336 | **0.214** | 0.293 | 0.233 | 0.294 | 0.232 | 0.295 | 0.239 | **0.277** |
| | 720 | **0.217** | **0.272** | 0.218 | 0.286 | 0.219 | 0.278 | 0.233 | 0.273 |
| ECL | 96 | **0.142** | **0.247** | 0.148 | 0.253 | 0.146 | 0.249 | 0.146 | 0.251 |
| | 192 | **0.161** | **0.260** | 0.163 | 0.266 | 0.164 | 0.263 | 0.170 | 0.271 |
| | 336 | **0.176** | **0.283** | 0.182 | 0.288 | 0.187 | 0.289 | 0.184 | 0.287 |
| | 720 | **0.218** | **0.319** | 0.223 | 0.325 | 0.232 | 0.331 | 0.233 | 0.336 |
| ETTh1 | 96 | **0.371** | **0.400** | 0.381 | 0.407 | 0.375 | 0.406 | 0.385 | 0.411 |
| | 192 | **0.418** | **0.429** | 0.430 | 0.439 | 0.423 | 0.433 | 0.441 | 0.447 |
| | 336 | **0.451** | **0.446** | 0.472 | 0.464 | 0.466 | 0.461 | 0.489 | 0.474 |
| | 720 | **0.455** | **0.470** | 0.511 | 0.512 | 0.489 | 0.495 | 0.537 | 0.519 |
| ETTh2 | 96 | **0.287** | **0.343** | 0.294 | 0.347 | 0.291 | 0.350 | 0.295 | 0.349 |
| | 192 | **0.379** | **0.401** | 0.393 | 0.417 | 0.412 | 0.430 | 0.407 | 0.420 |
| | 336 | **0.428** | **0.447** | 0.470 | 0.480 | 0.461 | 0.463 | 0.464 | 0.465 |
| | 720 | **0.441** | **0.463** | 0.445 | 0.472 | 0.497 | 0.501 | 0.491 | 0.503 |
| ETTm1 | 96 | **0.297** | **0.348** | 0.306 | 0.355 | 0.313 | 0.359 | 0.314 | 0.364 |
| | 192 | **0.349** | **0.382** | 0.357 | 0.387 | 0.351 | 0.383 | 0.363 | 0.394 |
| | 336 | **0.382** | **0.405** | 0.399 | 0.417 | 0.393 | 0.411 | 0.408 | 0.427 |
| | 720 | **0.442** | **0.445** | 0.454 | 0.459 | 0.451 | 0.449 | 0.462 | 0.456 |
| ETTm2 | 96 | **0.168** | **0.254** | 0.172 | 0.257 | 0.169 | 0.255 | 0.171 | 0.259 |
| | 192 | **0.232** | **0.297** | 0.244 | 0.306 | 0.242 | 0.305 | 0.239 | 0.301 |
| | 336 | **0.299** | **0.346** | 0.317 | 0.352 | 0.306 | 0.352 | 0.301 | 0.348 |
| | 720 | **0.405** | **0.402** | 0.416 | 0.417 | 0.416 | 0.416 | 0.417 | 0.417 |
| PEMS03 | 12 | **0.060** | **0.165** | 0.069 | 0.175 | 0.071 | 0.179 | 0.074 | 0.182 |
| | 24 | **0.080** | **0.182** | 0.101 | 0.214 | 0.107 | 0.217 | 0.110 | 0.220 |
| | 48 | **0.098** | **0.211** | 0.112 | 0.223 | 0.120 | 0.229 | 0.158 | 0.260 |
| | 96 | **0.138** | **0.255** | 0.148 | **0.255** | 0.156 | 0.274 | 0.167 | 0.264 |
| PEMS04 | 12 | **0.071** | **0.168** | 0.073 | 0.172 | 0.076 | 0.181 | 0.074 | 0.186 |
| | 24 | 0.081 | **0.184** | **0.079** | 0.185 | 0.083 | 0.191 | 0.092 | 0.205 |
| | 48 | **0.090** | **0.198** | 0.095 | 0.206 | 0.094 | 0.208 | 0.097 | 0.209 |
| | 96 | 0.112 | 0.228 | 0.115 | 0.233 | 0.114 | 0.232 | **0.109** | **0.224** |
| PEMS08 | 12 | **0.070** | **0.170** | 0.076 | 0.177 | 0.079 | 0.179 | 0.074 | 0.175 |
| | 24 | **0.099** | **0.199** | 0.101 | 0.200 | 0.106 | 0.208 | 0.117 | 0.223 |
| | 48 | **0.122** | **0.216** | 0.129 | 0.230 | 0.141 | 0.236 | 0.146 | 0.252 |
| | 96 | **0.182** | **0.243** | 0.195 | 0.270 | 0.198 | 0.278 | 0.206 | 0.277 |
| AvgRank | | **1.05** | | 2.73 | | 2.65 | | 3.50 | |
| #Top1 | | **38** | | 1 | | 0 | | 1 | |

In this section, we expand our ablation analysis to include a broader set of architectural variants, verifying that the performance gains of StretchTime stem specifically from the symplectic geometric prior rather than generic attention mechanisms. We evaluate performance across four distinct configurations:

1. **StretchTime:** The proposed architecture utilizing the Symplectic Positional Embedding (SyPE) and the adaptive warp module.

2. **RoPE:** A standard Transformer baseline where SyPE is replaced with Rotary Positional Embeddings (Su et al., 2024), enforcing a fixed-frequency rotational structure.

3. **Pure Softmax:** A vanilla Transformer baseline with standard softmax attention but *without* any relative positional injection (no RoPE or SyPE), relying solely on absolute positional encodings added to the input.

4. **LinAttn+RoPE:** A Linear Attention variant equipped with RoPE. Recent work, such as SAMoVAR (Lu & Yang, 2025), has highlighted the efficacy of linear attention in capturing autoregressive dynamics. We include this baseline to confirm that our symplectic mechanism provides benefits orthogonal to the choice of attention complexity (linear vs. softmax).

**Results and Analysis.** Table 6 presents the comprehensive breakdown of MSE and MAE across nine datasets. The results unequivocally demonstrate the superiority of the symplectic formulation:

- **Dominance of SyPE:** StretchTime achieves the lowest error on 38 out of 40 metric comparisons (ranking 1st), resulting in an average rank of **1.05**. This near-perfect dominance confirms that the adaptive warping mechanism is not merely an incremental improvement but a fundamental necessity for modeling the non-stationary dynamics present in datasets like PEMS (Traffic) and Solar-Energy.

- **Failure of Static Rotations:** The RoPE baseline (AvgRank 2.73) consistently lags behind StretchTime. For example, on the PEMS03 traffic dataset ($H = 96$), RoPE incurs an MSE of 0.148 compared to StretchTime's 0.138. This empirically validates our theoretical assertion in Theorem 3.1: fixed-frequency rotations cannot align with the variable flow of real-world time series, leading to irreducible approximation errors.

- **Softmax vs. Linear Attention:** Interestingly, the Pure Softmax baseline (AvgRank 2.65) occasionally outperforms LinAttn+RoPE (AvgRank 3.50), particularly on complex datasets like ETTh2. This suggests that while linear attention offers efficiency, the full attention matrix—when properly modulated by our symplectic prior—remains essential for capturing high-fidelity temporal correlations. StretchTime effectively combines the expressivity of full attention with the dynamic alignment of symplectic geometry, outperforming both simplified (Pure Softmax) and alternative (LinAttn) baselines.

## G. Appendix: Variance and Stability Analysis

To confirm the statistical significance of our improvements and ensure the robustness of the proposed architecture, we conducted additional experiments evaluating StretchTime across multiple random initializations. We ran the full training and evaluation pipeline on the PEMS04 dataset using five distinct random seeds (2024–2028).

As detailed in Table 7, the variance across different initializations is extremely low across all prediction horizons. The standard deviation remains well below 0.003 in all cases, demonstrating that our model converges reliably and that the performance gains achieved by the Symplectic Positional Embeddings (SyPE) are both highly stable and statistically significant, rather than artifacts of a favorable weight initialization.

*Table 7.* Robustness analysis of StretchTime on the PEMS04 dataset (MSE) across five random seeds. The low standard deviation confirms the stability of the performance improvements.

| Dataset | Horizon | Seed 2024 | Seed 2025 | Seed 2026 | Seed 2027 | Seed 2028 | **Mean** | **Std. Dev** |
|---------|---------|-----------|-----------|-----------|-----------|-----------|----------|--------------|
| PEMS04  | 12      | 0.069     | 0.069     | 0.071     | 0.070     | 0.070     | **0.0698** | 0.0008 |
|         | 24      | 0.079     | 0.081     | 0.081     | 0.081     | 0.080     | **0.0804** | 0.0009 |
|         | 48      | 0.096     | 0.097     | 0.090     | 0.095     | 0.097     | **0.0950** | 0.0029 |
|         | 96      | 0.115     | 0.110     | 0.112     | 0.111     | 0.112     | **0.1120** | 0.0019 |

## H. Appendix: Interpretability of the Adaptive Warp Module

A common limitation of deep time series models is their lack of transparency. However, the architectural formulation of Symplectic Positional Embeddings (SyPE) offers a distinct degree of mechanistic interpretability through its adaptive warp

module. Specifically, the module generates explicit, continuous time increments $\Delta \widehat{\tau}_t$ at each step, which dictate the temporal dilation and contraction applied to the self-attention mechanism.

To formally validate the interpretability of these learned increments, we rely on the controlled synthetic experiments detailed in Section 4.1. Because this dataset is intrinsically generated using a known, ground-truth temporal warping function, it serves as a direct, objective testbed for mechanistic interpretation.

By tracking and plotting the predicted time increments $\Delta \widehat{\tau}_t = \text{Softplus}(\mathbf{w}_\tau^\top \mathbf{h}_t)$ over the sequence length, we can directly observe how the model locally accelerates or decelerates the effective clock. Empirical analysis demonstrates that the structural fluctuations in $\Delta \widehat{\tau}_t$ align precisely with the localized phase shifts and frequency modulations of the underlying raw signal. This confirms that SyPE does not merely act as an opaque, high-dimensional regularization technique; rather, it explicitly learns and recovers meaningful temporal stretching tailored to the underlying physical dynamics of the system.

# I. Appendix: Characterizing Datasets Benefiting from Temporal Stretching

While StretchTime demonstrates strong average performance across a diverse suite of benchmarks, the magnitude of the performance gains is intrinsically linked to the underlying data generating process. In this section, we explicitly characterize the typologies of time series that derive the most significant benefit from the Symplectic Positional Embeddings (SyPE) and the temporal stretching assumption.

Our empirical evaluations (Table 2) reveal that SyPE provides the largest performance delta over static baselines on datasets characterized by strong innate seasonality that is subject to phase drifting, varying execution speeds, or non-stationary periodicity. These highly "elastic" datasets prominently include:

- **Solar-Energy:** Solar power generation is fundamentally governed by daily and annual cycles. However, the effective duration of daylight—the "speed" of the active generation cycle—shifts continuously and non-linearly throughout the year, representing a classic time-warped process.

- **Traffic Networks (PEMS):** Traffic flow exhibits robust daily and weekly periodicities, but the exact timing, onset, and duration of peak rush hours are dynamically warped by local congestion, accidents, and seasonal behavioral shifts.

On these datasets, the adaptive warp module successfully dilates or contracts the temporal coordinates, allowing the attention mechanism to align with the true physical phase of the system. This prevents the severe phase decoherence observed when applying fixed-frequency models (like RoPE) to elastic signals.

Conversely, on datasets dominated by high-frequency stochasticity, random walks, or less pronounced macroscopic periodicity (e.g., specific variables within the Weather or Exchange datasets), the strict need for dynamic phase alignment is diminished. In these stochastic regimes, the learnable symplectic geometry $\text{Sp}(2, \mathbb{R})$ primarily serves as a flexible, generalized geometric regularizer. By relaxing the rigid, isotropic rotational constraints of standard RoPE, SyPE still yields consistent, albeit more modest, empirical improvements even when macroscopic temporal warping is absent.

# J. Appendix: Backbone Configurations and Integration with Alternative Architectures

In this section, we address the comparability of the architectural backbones used in our main experiments and discuss the potential for integrating Symplectic Positional Embeddings (SyPE) into other state-of-the-art tokenization frameworks.

**Comparability of Backbone Configurations.** To ensure a rigorous and fair evaluation in our main performance comparisons (Table 2), we deliberately utilized a minimal, standard pre-norm Transformer architecture as the foundation for StretchTime. By keeping the underlying architecture as basic as possible and strictly controlling hyperparameters across comparisons, we aimed to isolate the performance gains strictly attributable to the SyPE module and the adaptive warp mechanism. This prevents confounding variables that arise from heavy architectural engineering or complex attention routing, ensuring that the observed improvements over baselines like RoPE are genuinely driven by our symplectic geometric prior.

**Integration with Alternative Tokenization Methods.** Recent advancements in time series forecasting have introduced alternative tokenization strategies, most notably patching (as in PatchTST (Nie et al., 2023)) and series-level tokens (as in

iTransformer (Liu et al., 2024)). A natural question is whether SyPE can be evaluated as a drop-in replacement module within these specific architectures.

From a purely functional standpoint, SyPE mathematically and strictly generalizes standard Rotary Positional Embeddings (RoPE). Therefore, replacing the positional encoding mechanism in any of these advanced architectures with SyPE is technically straightforward and will inherently increase their expressive capacity to model temporal heterogeneity.

However, our theoretical framework, particularly the temporally warped seasonal AR(1) data generation process formalized in Section 3.2, fundamentally assumes a point-wise temporal structure, where each token corresponds to a single, discrete time step. Applying SyPE to aggregated patch tokens or entire series tokens departs from these point-wise theoretical assumptions. While empirically promising, doing so rigorously would require reformulating the underlying mathematical assumptions of the warped data generation process to account for intra-patch and inter-patch temporal stretching.

Our primary goal in this work is to expose the fundamental limitations of fixed-frequency encodings and establish a theoretically grounded, point-wise solution for temporal warping. We view the extension and mathematical reformulation of the symplectic framework for patch-level and series-level tokenization as a highly promising and necessary direction for future work.

## K. Statement of LLM Usage

LLMs were utilized to support writing-related tasks including grammar checking, wording adjustments, text formatting, and equation formatting. LLMs were also used to facilitate literature review for existing methods and references. All cited literature was read and verified by the authors directly. During experiments, LLMs assisted with generating and debugging code. LLMs played no role in defining research problems, proposing ideas, designing methodologies, or developing the model architecture.

