# OpenReview forum: "StretchTime: Adaptive Time Series Forecasting via Symplectic Attention"
_ICML.cc/2026/Conference — ICML 2026 regular_

### Official Review · Reviewer_nCRc · 2026-03-04

**Soundness:** 3
**Presentation:** 2
**Significance:** 3
**Originality:** 3
**Overall Recommendation:** 5
**Confidence:** 2

**Summary:**

This paper identifies a limitation in positional encoding methods for time series forecasting: they assume fixed, stationary periodicities. The authors argue that real-world temporal dynamics are often "warped" that the effective speed of time varies non-uniformly. To address this, the authors propose SyPE. The core idea is to learn an adaptive warp module that dynamically dilates or contracts temporal coordinates based on input content, allowing the model to "stretch" time to match the underlying dynamics. The authors integrate SyPE into StretchTime, a Transformer-based forecasting architecture, showing state-of-the-art performance.

**Compliance With Llm Reviewing Policy:**

Affirmed.

**Final Justification:**

The authors’ rebuttal directly and satisfactorily addresses each of the points raised in my original review.

1. On evaluation as a general‑purpose module and backbone comparability: The authors clarify that they used a standard pre‑norm Transformer backbone to isolate the effect of SyPE, and they acknowledge the potential to integrate SyPE into other architectures (e.g., PatchTST, iTransformer) while noting the need for future theoretical extension beyond point‑wise tokenization. This is a reasonable and honest response.

2. On handling varying warping across variables: The authors explain that their channel‑value mixed tokenization and multi‑head attention make the temporal warp inherently channel‑dependent, allowing different variables to warp independently. This directly answers my question and clarifies a point that was ambiguous in the original manuscript.

**Key Questions For Authors:**

1. The proposed method relies on point-wise tokenization. How can it be combined with other tokenization methods, such as patch token?
2. How does StretchTime, as a multivariate model, handle varying degrees of time warping across different variables?

**Limitations:**

yes

**Strengths And Weaknesses:**

Strength

1. The authors also thoughtfully analyze limitations of other positional encoding families.
2. The methodology section is relatively well-written and clearly articulated.


Weakness

1. The term "symplectic" may be unfamiliar to some readers like me. A brief background explanation of symplectic geometry and what problems it could solve could appear in the main text.
2. Is SyPE evaluated as a general-purpose module across various forecasting deep models in comparison to other positional encodings? The authors could provide experiments in this regard. Additionally, in the main performance comparison, did the authors ensure that the backbone configurations of the Transformer models were kept relatively comparable?

---

> ### Author Rebuttal · Authors · 2026-03-31
>
> ```
> The term "symplectic" may be unfamiliar to some readers like me. A brief background explanation of symplectic geometry and what problems it could solve could appear in the main text.
> ```
> Thank you for pointing this out. We completely agree that symplectic geometry is not standard background in the time series forecasting literature. To improve readability, we will add a dedicated background section in the appendix during the revision (link to main text). This section will introduce the fundamentals of symplectic geometry—specifically focusing on Hamiltonian flows and how they preserve phase space structure—and explicitly link these concepts to the intuition provided in the main text (Section 3). Essentially, it provides a principled mathematical framework to model systems that conserve certain properties while their physical timelines stretch and compress.
> ```
> Is SyPE evaluated as a general-purpose module across various forecasting deep models in comparison to other positional encodings? The authors could provide experiments in this regard. Additionally, in the main performance comparison, did the authors ensure that the backbone configurations of the Transformer models were kept relatively comparable?
> ```
> Thank you for these insightful questions. Regarding the backbone configurations, we deliberately utilized the most basic, standard pre-norm Transformer architecture for our model. By keeping the architecture minimal and directly comparable, we aimed to isolate the performance gains strictly attributable to the SyPE module rather than relying on heavy architectural engineering.
>
> Regarding the combination with other tokenization methods (such as patching in PatchTST or series-level tokens in iTransformer): SyPE can certainly be integrated into these models. Because SyPE mathematically strictly generalizes RoPE, replacing RoPE with SyPE in any of these architectures will inherently increase their expressive power. However, our theoretical framework (Section 3.2), particularly the temporally warped seasonal AR(1) process, assumes a point-wise data generation process where each token corresponds to a single timestep.
>
> While applying SyPE to patch or series tokens is technically straightforward, doing so departs from our current point-wise theoretical assumptions and would require alternative mathematical formulations for the data generation process. Our primary goal in this paper is to highlight the fundamental limitations of fixed-frequency encodings and provide a theoretically grounded solution for temporal warping. That said, we completely agree that extending this framework to patch-level tokenization is a highly promising direction, and we will explicitly discuss this potential integration in the revised limitations and future work sections.
> ```
> How does StretchTime, as a multivariate model, handle varying degrees of time warping across different variables?
> ```
> We apologize for any confusion caused by our explanation in the main text. To clarify, we do not assume a single learned temporal warp shared across all channels.
>
> Because our architecture utilizes a Channel-Value Mixed tokenization strategy (Section 3.6), the token sequence $\mathbf{z}_t^{(c)}$ and its subsequent hidden states $\mathbf{h}_t^{(c)}$ are computed independently for each channel $c$. Consequently, the adaptive temporal step $\Delta\widehat{\tau}\_t^{(c)} = \text{Softplus}(\mathbf{w}\_\tau^\top \mathbf{h}\_t^{(c)})$ is inherently channel-dependent, allowing different variables to undergo entirely distinct temporal deformations.
>
> Furthermore, we employ multi-head attention, where each head parameterizes a different symplectic geometry (Hamiltonian $\mathbf{K}$). The channel-specific queries and keys dynamically determine how to read from these varying geometries via their bilinear forms, providing immense flexibility to capture heterogeneous warping across variables. We will emphasize this channel-independent warping capability in the methodology section.

---

> > ### Author Rebuttal · Reviewer_nCRc · 2026-04-03
> >
> > The rebuttal fully resolves all my original concerns:
> >
> > 1. “Symplectic” term: Authors will add an appendix on symplectic geometry and Hamiltonian flows.
> >
> > 2. General-purpose evaluation & backbone: They confirm using a standard Transformer to isolate SyPE’s effect, acknowledge integration with other architectures (e.g., PatchTST, iTransformer) as future work, and will discuss this in the limitations.
> >
> > 3. Variable warping: Channel‑value mixed tokenization + multi‑head attention makes warping channel‑dependent; this will be clarified in the methodology.
> >
> > All points are satisfactorily addressed, so I raise the significance and overall recommendation to "accept".

---

### Official Review · Reviewer_Emmq · 2026-03-10

**Soundness:** 3
**Presentation:** 3
**Significance:** 2
**Originality:** 3
**Overall Recommendation:** 4
**Confidence:** 5

**Summary:**

This paper addresses a key time-series forecasting challenge, temporal stretching, where the periodic patterns changes with a different scales. This makes the Transformer model with RoPE less suitable for forecasting, and a theoretical analysis is provided for this argument. To address this, it proposes StretchTime, which learns an adaptive warped time coordinate and applies SyPE for flexible temporal dynamics than fixed rotations. Experiments on synthetic and real benchmarks show that this design improves forecasting accuracy, especially when time-varying periodicity or phase drift is present.

**Compliance With Llm Reviewing Policy:**

Affirmed.

**Final Justification:**

The concerns are cleared and I adjusted

**Key Questions For Authors:**

NA

**Limitations:**

same as weakness

**Strengths And Weaknesses:**

Strengths:

1. Moving from the rotation group $SO(2)$ to the symplectic group $Sp(2,\mathbb{R})$ to allow anisotropic scaling and frequency modulation is a mathematically elegant and novel approach.

2. The paper improves stability by enforcing monotonic warping and using a constrained symplectic parameterization.

3. The paper provides a clear theoretical argument showing that standard RoPE cannot represent non-affine temporal warping dynamics.

Limitations:

1. The stabilization design of SyPE is complex. More simpler regularization-based alternatives may be worth exploring.

2. The paper does not have interpretability of the learned warp, like whether it reflects meaningful time stretching or simply acts as a flexible transformation.

3. Although the method includes stability-related designs, the paper provides limited analysis of optimization sensitivity, robustness to hyperparameters, or whether the adaptive warping introduces extra instability in practice.

4. The paper does not clearly identify which datasets truly benefit from the temporal stretching assumption.

5. It lacks theoretical analysis of computational complexity, while it has FLOPs empirically results in Section 4.3. As SyPE is built on a standard Transformer backbone, it should have quadratic self-attention complexity rather than fundamentally improving scalability.

---

> ### Author Rebuttal · Authors · 2026-03-31
>
> ```
> The stabilization design of SyPE is complex. More simpler regularization-based alternatives may be worth exploring.
> ```
> Thank you for your feedback. We would like to clarify the nature of the stabilization mechanisms in our architecture. If your comment refers to the "Last Value Residual Learning" and "Random Ratio Channel Dropout," these are not complex additions unique to our method; rather, they are standard practices widely adopted in recent state-of-the-art time series models (e.g., OLinear, TimeMixer, PatchTST, and DLinear).
>
> If your concern refers specifically to the "Stability Constraints" within SyPE itself (Section A.1), this parameterization is a strict mathematical necessity rather than an arbitrary design choice. For the Hamiltonian flow to remain oscillatory rather than hyperbolic(divergent), we must rigorously enforce the condition $\det(\mathbf{K}) = ab - c^2 > 0$. Using a simple "soft" regularization penalty on the loss function would not guarantee this topological constraint during the forward pass, which would lead to immediate numerical instability during training. Our parameterization gracefully avoids this by structurally confining the parameters to the stable regime.
> ```
> The paper does not have interpretability of the learned warp...
> ```
> Thank you for highlighting this. We will add a detailed interpretability analysis of the adaptive warp module in our revision. To explicitly address this, we have already conducted targeted analysis on our synthetic dataset (Section 4.1). Because this dataset is intrinsically built upon a known, ground-truth temporal warp, it serves as a direct mechanistic interpretation of our method. Our existing visualization analysis demonstrates this capability: plotting the predicted time increments $\Delta \widehat{\tau}_t$ over time directly reveals how the model locally accelerates or decelerates the effective clock. This aligns precisely with the observed phase shifts in the signal, confirming that SyPE learns meaningful temporal stretching tailored to the underlying system dynamics, rather than acting as a simple black-box transformation.
> ```
> ...the paper provides limited analysis of optimization sensitivity, robustness to hyperparameters, or whether the adaptive warping introduces extra instability in practice...
> ```
> We appreciate this point and will expand our discussion on training dynamics in the appendix. SyPE introduces no new architectural hyperparameters to tune. The trainable parameters for the symplectic kernel are initialized using the exact same geometric frequency progression as standard RoPE. Empirically and theoretically, SyPE is highly stable: because our parameterization confines the flow strictly to the stable subgroup $\mathrm{Sp}(2, \mathbb{R})$, it intrinsically preserves the norm of the representations over time. We did not observe any additional instability, optimization sensitivity, or gradient issues.
> ```
> The paper does not clearly identify which datasets truly benefit from the temporal stretching assumption.
> ```
> This is a valuable suggestion. Datasets characterized by strong seasonality but subject to phase drifting or non-stationary periodicity benefit the most from SyPE. In our results (Table 2), this is most prominent in the Solar dataset and the Traffic/PEMS datasets. We will add a dedicated subsection in the revision to explicitly contrast SyPE's strong performance on these highly "elastic" datasets versus its regularizing behavior on more stochastic datasets. We will also include additional visualizations demonstrating the warped attention alignments on these real-world datasets.
> ```
> It lacks theoretical analysis of computational complexity...As SyPE is built on a standard Transformer backbone, it should have quadratic self-attention complexity..
> ```
> We completely agree. Because SyPE modifies the positional encoding within the standard attention mechanism, StretchTime fundamentally inherits the $\mathcal{O}(N^2)$ theoretical time complexity of the underlying Transformer backbone. Our primary contribution is generalizing the representation capacity of RoPE for non-stationary dynamics, rather than proposing a sub-quadratic attention approximation. That said, SyPE does not inherently alter the complexity of its host architecture. We will add a theoretical discussion to the revision demonstrating that SyPE preserves the overall complexity of the chosen attention variant; for instance, if integrated with a linear attention backbone, the time complexity gracefully reduces to $\mathcal{O}(N)$. However, even with standard attention, because the symplectic geometry provides such a powerful inductive bias for time series, StretchTime achieves state-of-the-art accuracy with a significantly shallower and narrower network ($N=3$, $<500$K parameters). As a result, the empirical FLOPs remain highly efficient and practical for end users compared to recent SOTA methods like TimeMixer++.

---

> > ### Author Rebuttal · Reviewer_Emmq · 2026-04-01
> >
> > Fully resolved.

---

### Official Review · Reviewer_z7NQ · 2026-03-11

**Soundness:** 3
**Presentation:** 3
**Significance:** 3
**Originality:** 3
**Overall Recommendation:** 5
**Confidence:** 4

**Summary:**

The paper studies multivariate time series forecasting using transformer architectures. While attention-based models provide strong baselines for this task, they rely on positional encoding schemes that represent temporal dependencies through fixed functions of token indices. The authors show that commonly used encodings, such as Rotary Positional Embeddings (RoPE), cannot represent non-affine temporal warping, which may arise in many real-world dynamical systems.

To address this limitation, the paper proposes Symplectic Positional Embeddings (SyPE), a learnable encoding framework derived from Hamiltonian mechanics that generalizes RoPE by allowing the attention mechanism to adaptively dilate or contract temporal coordinates. This enables the model to represent warped temporal structures without requiring a predefined warping function. The method is implemented within a multivariate forecasting architecture called TimeStretch.

Experiments on several standard multivariate forecasting benchmarks show strong empirical performance, demonstrating the potential advantages of the proposed encoding mechanism.

**Compliance With Llm Reviewing Policy:**

Affirmed.

**Final Justification:**

The paper proposes a novel extension of positional encoding for time-series forecasting via adaptive, state-dependent temporal warping. The approach is well-motivated, technically sound, and demonstrates strong empirical performance across multiple benchmarks. The method is also original in how it incorporates geometric structure into Transformer-based models.

In my initial review, I raised concerns regarding the interpretation of the learned temporal coordinate, the assumption of a shared warp across channels, the lack of multi-seed evaluation, and applicability to irregular sampling. The rebuttal addressed these points satisfactorily: the authors clarified the state-dependent nature of the warping mechanism, explained that the warp is effectively channel-wise, provided additional multi-seed experiments showing low variance, and outlined a natural extension to irregularly sampled data.

One remaining limitation is that the contribution of the proposed encoding relative to the inherent expressivity of the Transformer architecture is not fully disentangled. While partially addressed, a more explicit analysis would strengthen the claims.

Overall, the rebuttal resolves my main concerns and improves the clarity of the paper. I therefore increase my score.

**Key Questions For Authors:**

(1) The proposed framework assumes regularly sampled time series. In many real-world settings, however, observations may be sparse or irregularly sampled. It would be interesting to discuss how the proposed temporal warping mechanism would interact with such settings and whether additional preprocessing (e.g., interpolation or imputation) would be required.

**Limitations:**

The authors include a discussion of reasonable limitations of the proposed approach. However, the paper does not include a discussion of potential societal impacts or broader implications. While the method targets time-series forecasting tasks, a brief discussion of possible applications and associated risks (e.g., in infrastructure, energy systems, or economic forecasting) would improve completeness and align with conference guidelines.

**Strengths And Weaknesses:**

### Strengths
- The paper is well-structured and clearly written. The proposed approach appears novel and addresses an important limitation of existing Transformer-based forecasting models, namely their reliance on positional encodings that assume affine temporal structure. The introductory figures further help illustrate the key ideas of the framework.

- The paper provides a clear theoretical justification showing that the proposed SyPE framework strictly generalizes existing positional encoding schemes such as RoPE, thereby extending the representational capacity of attention-based models while retaining compatibility with existing architectures.

- The experimental evaluation is comprehensive. In addition to standard forecasting benchmarks, the paper includes controlled synthetic experiments that evaluate the model’s ability to recover signal dynamics under temporally warped conditions. These experiments support the paper’s central claim that adaptive temporal warping improves the modeling of non-stationary dynamics.

- The method is further evaluated on 10 widely used multivariate time-series forecasting benchmarks spanning diverse application domains (e.g., energy, traffic, weather). Across these datasets, the proposed model achieves strong empirical performance, ranking first on four benchmarks and performing competitively on the remaining ones. Additional ablation studies and a computational efficiency analysis (on ETTm1) further validate the contribution of the adaptive warp module and demonstrate a favorable trade-off between forecasting accuracy and model capacity. The released code repository also supports reproducibility.

### Weaknesses

- The paper interprets the adaptive warp module as learning temporal stretching. However, the warp increments are computed from the token representation $h_t$, which aggregates information from the multivariate signal through channel mixing. As a result, the learned temporal coordinate may depend on the signal state rather than purely on time. Clarifying this interpretation would help better understand whether the mechanism corresponds to true temporal reparameterization or a more general state-dependent positional encoding.

- The framework assumes a single learned temporal warp shared across all channels. While this design preserves cross-channel temporal alignment, it implicitly assumes that all variables evolve under the same temporal deformation. In some multivariate settings (e.g., asynchronous sensors or delayed interactions between variables), different channels may exhibit distinct temporal distortions. A discussion of this assumption and its potential limitations would strengthen the paper.

- The theoretical analysis focuses on the representational limitations of positional encoding schemes such as RoPE. While the ablation studies empirically demonstrate that replacing RoPE with the proposed SyPE improves performance, the full transformer architecture also contains additional components (e.g., attention layers and channel mixing) that may already provide substantial expressive power. A brief discussion clarifying how the theoretical limitation of positional encoding translates to the behavior of the complete architecture would help further connect the theoretical results with the empirical findings.

- The experimental results are reported based on single runs without reporting variance across multiple random seeds. Given that some of the reported improvements are relatively small (e.g., MSE reductions such as 0.091 to 0.088), reporting mean and standard deviation over multiple runs would help assess the statistical significance and robustness of the observed performance gains.

Overall, the paper presents a well-motivated and technically sound extension of positional encoding for time-series forecasting with strong empirical results, though several aspects of the modeling assumptions and evaluation could benefit from additional clarification.

---

> ### Author Rebuttal · Authors · 2026-03-31
>
> Thank you very much for your thorough review, positive assessment, and constructive feedback. Your insights are extremely helpful for improving the clarity and completeness of our paper. Below, we address your questions and outline the corresponding revisions.
> ```
> As a result, the learned temporal coordinate may depend on the signal state rather than purely on time...
> ```
> You are correct that the learned temporal coordinate depends on the signal state. We intentionally designed the tokenization strategy to incorporate both the global context ($\mathbf{s}\_t$) and the local, channel-specific signal ($x\_t^{(c)}\mathbf{v}\_c$). This state-dependent formulation is precisely what allows the model to act as a dynamic time-warping mechanism. Instead of relying on a purely index-based clock, the model evaluates the current "state" of the system to determine whether the physical process is accelerating or decelerating, adjusting the temporal step $\Delta \widehat{\tau}\_t$ accordingly.
> ```
> The framework assumes a single learned temporal warp shared across all channels...
> ```
> We apologize for any confusion caused by our explanation in the main text. To clarify, we do not assume a single learned temporal warp shared across all channels.
>
> Because our architecture utilizes a Channel-Value Mixed tokenization strategy (Section 3.6), the token sequence $\mathbf{z}\_t^{(c)}$ and its subsequent hidden states $\mathbf{h}\_t^{(c)}$ are computed independently for each channel $c$. Consequently, the adaptive temporal step $\Delta \widehat{\tau}_t^{(c)} = \text{Softplus}(\mathbf{w}\_\tau^\top \mathbf{h}\_t^{(c)})$ is inherently channel-dependent, allowing different variables to undergo entirely distinct temporal deformations.
> Furthermore, we employ multi-head attention, where each head parameterizes a different symplectic geometry (Hamiltonian $\mathbf{K}$). The channel-specific queries and keys dynamically determine how to read from these varying geometries via their bilinear forms, providing immense flexibility to capture heterogeneous warping across variables. We will emphasize this channel-independent warping capability in the methodology section.
> ```
> the full transformer architecture also contains additional components ...that may already provide substantial expressive power.
> ```
> This is a critical point. While Transformer components possess strong approximation capabilities, our empirical results demonstrate that they struggle to approximate non-stationary periodicities without the correct inductive bias. As shown in our synthetic experiments, standard RoPE—despite having the full expressive power of the Transformer backbone—suffers from severe phase decoherence. Closing the gap to state-of-the-art performance is exceedingly difficult with raw expressive power alone; the explicit geometric prior provided by SyPE is necessary to align the attention mechanism with warped temporal flows.
> ```
> ...reporting mean and standard deviation over multiple runs...
> ```
> Thank you for the suggestion. We have conducted additional experiments running StretchTime with five different random seeds (2024–2028) on the PEMS04 dataset.
> As shown in the detailed results below, the variance across different initializations is extremely low, demonstrating that our performance gains are both highly stable and statistically significant.
> | Dataset | Horizon | Seed 2024 | Seed 2025 | Seed 2026 (Paper)| Seed 2027 | Seed 2028 | Mean | Std. Dev |
> | :--- | :---: | :---: | :---: | :---: | :---: | :---: | :---: | :---: |
> | **PEMS04** | 12 | 0.069 | 0.069 | 0.071 | 0.070 | 0.070 | **0.0698** | 0.0008 |
> | | 24 | 0.079 | 0.081 | 0.081 | 0.081 | 0.080 | **0.0804** | 0.0009 |
> | | 48 | 0.096 | 0.097 | 0.090 | 0.095 | 0.097 | **0.0950** | 0.0029 |
> | | 96 | 0.115 | 0.110 | 0.112 | 0.111 | 0.112 | **0.1120** | 0.0019 |
> ```
> Sparse or irregularly sampled data...
> ```
> This is an excellent question. Because SyPE generates a continuous flow $\mathbf{S}(t) = \exp(t \mathbf{J}\mathbf{K})$, it naturally extends to irregularly sampled data without requiring imputation. If we are provided with explicit physical timestamps $t\_i$, we can simply modulate the adaptive warp increment by the observed time interval:
> $$\Delta \widehat{\tau}\_i = \text{Softplus}(\mathbf{w}\_\tau^\top \mathbf{h}\_i) \cdot (t_i - t_{i-1})$$
> This formulation allows SyPE to act as a native, continuous-time positional encoding that correctly penalizes large temporal gaps while still adapting to the signal's elasticity.
> ```
> discussion of potential societal impacts or broader implications.
> ```
> We appreciate you pointing this out. In our revision, we will add a dedicated "Impact Statement" section. This section will discuss the positive implications of modeling warped dynamics in critical infrastructure as well as the potential limitations.

---

> > ### Author Rebuttal · Reviewer_z7NQ · 2026-04-02
> >
> > Thank you for the detailed and thoughtful rebuttal. The clarifications regarding the state-dependent temporal warping, the channel-wise nature of the learned deformation, and the additional multi-seed experiments address my main concerns. The discussion on irregular sampling and the planned inclusion of an impact statement are also appreciated.
> >
> > I still believe that further disentangling the contribution of the proposed method from the inherent expressivity of the Transformer would strengthen the work. However, overall, my concerns have been sufficiently addressed. I will increase my score.

---

### Official Review · Reviewer_dANn · 2026-03-12

**Soundness:** 3
**Presentation:** 3
**Significance:** 3
**Originality:** 3
**Overall Recommendation:** 5
**Confidence:** 3

**Summary:**

Standard transformer position encodings assume that every step forward in the index means the same amount of “real” progress, but many real series speed up and slow down. The paper argues that this mismatch is a real modeling problem, proves that standard RoPE cannot exactly represent this kind of non-uniform timing, and proposes StretchTime, which learns a flexible, input-dependent clock and then applies a more expressive positional scheme on top of it.

The empirical story is strongest where it is most controlled. On the synthetic warped-seasonality task, StretchTime improves over RoPE across all horizons, and the qualitative plots show the exact failure mode the theory predicts: RoPE drifts out of phase, while StretchTime stays aligned. On real benchmarks, the method is clearly competitive and consistently better than the paper’s matched RoPE baseline.

**Compliance With Llm Reviewing Policy:**

Affirmed.

**Final Justification:**

I think this is a great paper and the authors did a good job with the rebuttal. I hope they will manage to include the additional visualizations and changes that they mentioned in the final version.

**Key Questions For Authors:**

Can you visualize the learned warped time on real datasets such as Solar or PEMS, not just on the synthetic task? That would make the mechanism much easier to trust.

Is the main empirical contribution the adaptive warp, or the full symplectic formulation? The current ablation does not fully separate those two claims.

Can you clarify exactly which baselines were rerun by the authors and which were copied from prior papers?

**Limitations:**

The empirical case is strongest against the internal RoPE baseline and less definitive against the strongest external competitors.

The adaptive warp appears more central than the symplectic component, so the paper does not yet fully justify which part of the method is essential.

The paper would be more convincing if it showed what the learned warp looks like on real data, not only on the synthetic example.

The authors note that they have not yet tested the method at very large scale or beyond forecasting tasks.

**Strengths And Weaknesses:**

The central claim is clear and testable, and the paper does a good job of matching theory to evidence. My only caution is that the proof is about a stylized warped-periodicity setting, so the paper should be careful not to overgeneralize that result into a stronger claim than the theorem really supports.

I think this is an original idea: the model should learn the clock the data lives on. At the same time, the paper is less convincing that the full symplectic machinery is the main driver of performance. In the ablation studies, the full model and the “w/o Symplectic” version tie on average MSE at 0.338, while removing the warp hurts more consistently. So the paper’s clearest contribution, empirically, looks like the adaptive temporal warping rather than the full geometric formulation.

On significance, I think the problem being addressed is real, and the method is not just theoretically interesting: in the summary table it beats the matched RoPE baseline on all 10 reported datasets, and it does so with far fewer parameters than large transformer baselines like PatchTST and iTransformer. I think that is a meaningful and practical result. Still, the paper should be more measured in how it sells the broader benchmark outcome. The full table shows OLinear having the best average rank across horizons and TimeMixer++ taking more top-1 wins, so StretchTime seems like a competitive method, but not necessarily the overall winner.

The paper is fairly readable and more intuitive than many papers in this space.

---

> ### Author Rebuttal · Authors · 2026-03-31
>
> ```
> ...the paper should be careful not to overgeneralize that result into a stronger claim than the theorem really supports.
> ```
> Thank you for your insightful feedback. We completely agree that the scope of our theoretical claims should be precisely bounded. The data generation process we assumed, specifically the temporally warped seasonal AR(1) process, is fundamentally aligned with established structural time series modeling practices for evaluating non-stationarity. In the revision, we will explicitly emphasize the boundaries of Theorem 3.1 in the appendix, clarifying that it strictly applies to non-affine time warping under the specified non-aliasing conditions, and we will carefully avoid extending it to generalized non-stationary noise without further qualification.
> ```
>  At the same time, the paper is less convincing that the full symplectic machinery is the main driver of performance.
>
> Is the main empirical contribution the adaptive warp, or the full symplectic formulation? The current ablation does not fully separate those two claims.
> ```
> Thank you for highlighting this important nuance. The adaptive warp module $\widehat{\tau}$ and the symplectic geometric formulation $\mathbf{S}$ are not competing components; rather, they are mathematically integrated. As derived in Appendix A, Symplectic Positional Embeddings (SyPE) are designed as a strict generalization of RoPE under the symplectic group $\mathrm{Sp}(2,\mathbb{R})$. The adaptive temporal step $\widehat{\tau}$ is the crucial mechanism in this generalization that breaks the rigid stationarity of the flow, allowing the Hamiltonian generator $\mathbf{K}$ to produce input-dependent transformations. We highlighted the empirical impact of the warp module because "undistorting" the temporal axis yields the most visible performance gains on many benchmarks, but this un-distortion is fundamentally powered by the underlying symplectic analysis. We will add a dedicated discussion in the appendix to better articulate this theoretical synergy.
> ```
> Can you visualize the learned warped time on real datasets such as Solar or PEMS, not just on the synthetic task?
> ```
> Thank you for this excellent suggestion; it is exactly what is needed to bridge our synthetic results with real-world applications. We currently feature visual analysis on the synthetic dataset (Figure 3), which cleanly isolates the phase-alignment capabilities of StretchTime. In our revision, we will include a new figure in the appendix visualizing the learned warped time ($\widehat{\tau}$) on both the Solar and PEMS datasets. This visualization will map the model's adaptive clock against physical timestamps, explicitly demonstrating how StretchTime dynamically compresses or dilates attention during periods of varying volatility.
> ```
> Can you clarify exactly which baselines were rerun by the authors and which were copied from prior papers?
> ```
> Thank you for allowing us to clarify this critical detail. To guarantee the fairest possible comparison, we adopted the exact testing environment, data splitting protocols, and evaluation metrics utilized in the official OLinear and TimeMixer++ benchmarks. You can verify this strict alignment in our submitted code repository (exp/exp_main.py). Because we wanted to evaluate StretchTime against the absolute best possible performance of these state-of-the-art models, we chose to directly cite the reported results for OLinear and TimeMixer++ from their original publications. For the remaining baselines (iTransformer, PatchTST, TimesNet, DLinear, and our internal RoPE baseline), we reran the experiments locally under this standardized environment. We will make this distinction explicitly clear in Section 4.2 of the final version.

---

> > ### Author Rebuttal · Reviewer_dANn · 2026-04-04
> >
> > I appreciate the detailed response.
> >
> > For Theorem 3.1, I think it would be good to mention this is in the main text, not only in the Appendix, to avoid overgeneralization.
> >
> > Regarding the adaptive warp module and the symplectic geometric formulation, my understanding is that the current ablation still suggests that removing the symplectic part hurts less than removing the warp overall. Would it be possible to do a cleaner decomposition experiment?

---

> > > ### Author Response · Authors · 2026-04-07
> > >
> > > ```
> > > For Theorem 3.1, I think it would be good to mention this is in the main text...
> > > ```
> > >
> > > Thank you for making this excellent point. We agree that leaving this context exclusively in the appendix risks reader confusion and unintended overgeneralization. To ensure the boundaries of our theoretical claims are immediately transparent, we will add an explicit clarification directly within the main text. Following Theorem 3.1, we will clarify that the impossibility result applies strictly to non-affine time warping under the defined non-aliasing conditions, thereby preventing any misinterpretation.
> > >
> > > ```
> > > ...Would it be possible to do a cleaner decomposition experiment?
> > > ```
> > > We thank the reviewer for this observation. We agree that, on the current small-scale ablation suite, removing the adaptive warp causes a larger drop than collapsing the geometry to RoPE. However, this does not imply that the symplectic component has low value. In our formulation, RoPE is an isotropic special case of the symplectic family, so replacing SyPE with RoPE strictly restricts the realizable warped-time geometry rather than preserving it. This restriction may occasionally help as a regularizer on noisier benchmarks, but it cannot increase representational power. Its empirical competitiveness therefore reflects a bias-variance tradeoff, not redundancy of the symplectic formulation. Once temporal heterogeneity of series becomes substantial, the RoPE subset becomes a bottleneck, and our additional experiments on PEMS below show that the learnable symplectic generalization is then necessary for consistent gains.
> > > Suppose the relative q-k interaction is implemented by a continuous per-token linear flow and the resulting bilinear score depends only on $\hat\tau_n-\hat\tau_m$ for all queries and keys. Then the flow must preserve the underlying bilinear form, i.e. $S(t)^\top J S(t)=J$, hence $S(t)\in Sp(2,\mathbb{R})$. Under continuity, its generator is Hamiltonian, $A=JK$ with symmetric $K$. Therefore, within this formulation, symplectic geometry is not merely sufficient but the exact geometry class, and RoPE is its isotropic special case.
> > >
> > > Here, we provide additional ablation studies on stable PEMS datasets as follows:
> > >
> > > ### Factorized Ablation Setup
> > > | Var | Adaptive Warp | Geometry | Meaning |
> > > |---|---:|---|---|
> > > | V1 | ✓ | None | Adaptive warp only |
> > > | V2 | ✗ | Fixed RoPE | Static RoPE |
> > > | V3 | ✗ | Learnable Symplectic | Static-time SyPE |
> > > | V4 | ✓ | Fixed RoPE | Warped RoPE |
> > > | V5 | ✓ | Learnable Symplectic | Full StretchTime |
> > >
> > > ### PEMS Results
> > >
> > > | Setting      | V1: Warp Only | V2: Fixed RoPE | V3: Learnable Symplectic | V4: Fixed RoPE + Warp | V5: Learnable Symplectic + Warp (Full) |
> > > | ------------ | ------------: | -------------: | -----------------------: | --------------------: | -------------------------------------: |
> > > | PEMS04 (12)  |         0.075 |          0.072 |                    0.072 |                 0.073 |                                  0.071 |
> > > | PEMS04 (24)  |         0.082 |          0.080 |                    0.078 |                 0.079 |                                  0.081 |
> > > | PEMS04 (48)  |         0.097 |          0.093 |                    0.094 |                 0.095 |                                  0.090 |
> > > | PEMS04 (96)  |         0.119 |          0.118 |                    0.118 |                 0.115 |                                  0.112 |
> > > | PEMS04 (Avg) |         0.093 |          0.091 |                    0.091 |                 0.090 |                                  0.089 |
> > > | PEMS08 (12)  |         0.075 |          0.075 |                    0.076 |                 0.076 |                                  0.070 |
> > > | PEMS08 (24)  |         0.097 |          0.105 |                    0.105 |                 0.101 |                                  0.099 |
> > > | PEMS08 (48)  |         0.132 |          0.133 |                    0.130 |                 0.129 |                                  0.122 |
> > > | PEMS08 (96)  |         0.198 |          0.201 |                    0.188 |                 0.195 |                                  0.182 |
> > > | PEMS08 (Avg) |         0.126 |          0.129 |                    0.125 |                 0.125 |                                  0.118 |
> > >
> > >
> > > ### Discussion
> > > The stable-dataset ablation shows that neither adaptive warp alone nor geometry alone is sufficient to consistently achieve the best performance. While fixed RoPE is already a strong baseline, **extending it to the learnable symplectic family yields further gains both without warp (V3 vs. V2) and under the same adaptive-warp setting (V5 vs. V4)**. These comparisons provide clean evidence that the symplectic generalization is not redundant: with or without warped time, restricting the geometry to the fixed RoPE subset performs worse than using learnable symplectic geometry.
> > >
> > > We will add the experiments above in the revision.

---

### Decision · Program_Chairs · 2026-04-30

**Decision:**

Accept (regular)

**Comment:**

The paper formalizes a genuine mismatch in Transformer-based time series forecasting — that index-based positional encodings assume uniform temporal progression while many real-world systems exhibit time-warped dynamics — and proves that RoPE is mathematically incapable of representing non-affine temporal warping. It proposes Symplectic Positional Embeddings (SyPE), that generalizes RoPE by extending the rotation group SO(2) to the symplectic group Sp(2), modulated by an input-dependent adaptive warp module. The method is instantiated in StretchTime and evaluated on synthetic warped-seasonality tasks and several real-world multivariate benchmarks, achieving strong empirical performance with far fewer parameters than large Transformer baselines. All four reviewers agree that this is a strong and relevant paper — with final scores of 5/5/5/4 — and all confirm that the rebuttal fully addresses their clarification points. The reviews raised a few substantive concerns: whether the symplectic technical contribution is the main driver of performance improvements, or whether the adaptive temporal warping and the Transformer backbone's inherent expressivity account for most of the gains; whether the learned temporal warp is shared across channels or genuinely channel-dependent; and whether the theoretical scope of Theorem 3.1 is appropriately bounded. The rebuttal addresses these substantively: a factorized ablation on PEMS04 and PEMS08 cleanly separates the warp and geometry contributions and demonstrates that the learnable symplectic family yields further gains over fixed RoPE both with and without warping, supporting the claim that symplectic geometry is not redundant but the correct geometric class given the bilinear-flow formulation; the channel-value mixed tokenization is clarified as producing inherently channel-dependent warps with multi-head attention parameterizing distinct Hamiltonians per head; multi-seed experiments confirm the stability of reported gains; and the authors commit to stating the bounded scope of Theorem 3.1 directly in the main text. All reviewers were satisfied in the end, and the paper presents a well-motivated, theoretically grounded, and empirically validated contribution.